# Deep Neural Collapse Is Provably Optimal for the Deep Unconstrained Features Model

**Peter Súkeník**
Institute of Science and Technology
Klosterneuburg, Austria
peter.sukenik@ista.ac.at

**Marco Mondelli**[*]
Institute of Science and Technology
Klosterneuburg, Austria
marco.mondelli@ista.ac.at

**Christoph H. Lampert**[*]
Institute of Science and Technology
Klosterneuburg, Austria
chl@ist.ac.at

## Abstract

Neural collapse (NC) refers to the surprising structure of the last layer of deep neural networks in the terminal phase of gradient descent training. Recently, an increasing amount of experimental evidence has pointed to the propagation of NC to earlier layers of neural networks. However, while the NC in the last layer is well studied theoretically, much less is known about its multi-layered counterpart – deep neural collapse (DNC). In particular, existing work focuses either on linear layers or only on the last two layers at the price of an extra assumption. Our paper fills this gap by generalizing the established analytical framework for NC – the unconstrained features model – to multiple non-linear layers. Our key technical contribution is to show that, in a deep unconstrained features model, the unique global optimum for binary classification exhibits all the properties typical of DNC. This explains the existing experimental evidence of DNC. We also empirically show that *(i)* by optimizing deep unconstrained features models via gradient descent, the resulting solution agrees well with our theory, and *(ii)* trained networks recover the unconstrained features suitable for the occurrence of DNC, thus supporting the validity of this modeling principle.

## 1 Introduction

In the thought-provoking paper [22], Papyan *et al.* uncovered a remarkable structure in the last layer of sufficiently expressive and well-trained deep neural networks (DNNs). This phenomenon is dubbed "neural collapse" (NC), and it has been linked to robustness and generalization of DNNs. In particular, in [22] it is shown that all training samples from a single class have the same feature vectors after the penultimate layer (a property called NC1); the globally centered class-means form a *simplex equiangular tight frame*, the most "spread out" configuration of equally-sized vectors geometrically possible (NC2); the last layer's classifier vectors computing scores for each of the classes align with the centered class-means (NC3); finally, these three properties make the DNN's last layer act as a nearest class-center classifier on the training data (NC4).

The intriguing phenomenon of neural collapse has spurred a flurry of interest aimed at understanding its emergence, and the unconstrained features model (UFM) [21] has emerged as a widely accepted theoretical framework for its analysis. In this model, one assumes that the network possesses such an

---

[*]Equal contribution

37th Conference on Neural Information Processing Systems (NeurIPS 2023).

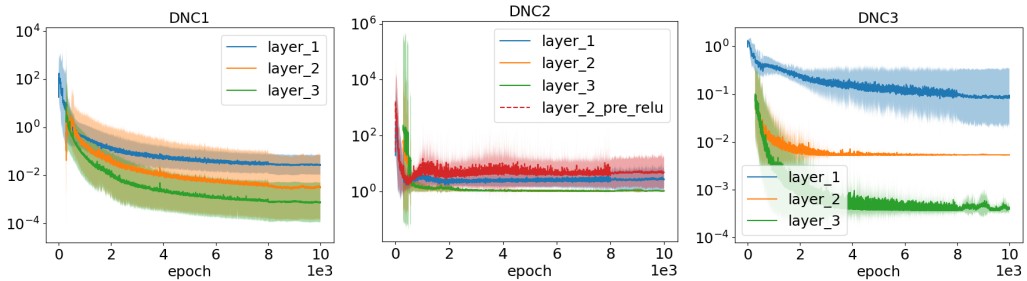

Figure 1: Deep neural collapse metrics as a function of training progression. On the left, DNC1 measures within-class variability collapse (the closer to 0, the more collapsed); in the middle, DNC2 measures closeness of feature activations to an orthogonal frame (the closer to 1, the more collapsed); and on the right, DNC3 measures alignment between feature activations and weights (the closer to 0, the more collapsed). The ResNet20 recovers the DNC metrics for DUFM, in accordance with our theory. This validates unconstrained features as a modeling principle for neural collapse.

expressive power that it can represent arbitrary features on the penultimate layer's output. Thus, the feature representations of the training samples on the penultimate layer are treated as free optimization variables, which are optimized for (instead of the previous layers' weights). Since its introduction, the UFM has been understood rather exhaustively, see e.g. [21, 19, 31, 3, 12, 42] and Section 2.

Most recently, researchers started to wonder, whether NC affects solely the last layer. The "cascading neural collapse" is mentioned in [11], where it is pointed out that NC1 might propagate to the layers before the last one. Subsequently, a mathematical model to study this deep version of NC has been proposed in [28], albeit only for two layers; and a deep model has been considered in [2], albeit only for linear activations. On the empirical side, the propagation of NC1 through the layers is demonstrated in [8], and the occurrence of all the NC properties in multiple layers is extensively measured in [26]. All these works agree that neural collapse occurs to some extent on the penultimate and earlier layers, leaving as a key open question the rigorous characterization of its emergence.

To address this question, we generalize the UFM and its extensions in [28, 2] to cover an arbitrary number of non-linear layers. We call this generalization the *deep unconstrained features model (DUFM)*. The features, as in the original UFM, are still treated as free optimization variables, but they are now followed by an arbitrary number of fully-connected layers with ReLU activations between them, which leads to a natural setup to investigate the theoretical underpinnings of deep neural collapse (DNC). Our main contribution is to prove (in Section 4) that DNC is the only globally optimal solution in the DUFM, for a binary classification problem with $l_2$ loss and regularization. This extends the results in [28] and [2] by having more than two layers and non-linearities, respectively. While the statements of [28] are formulated for an arbitrary number of classes, here we restrict ourselves to binary classification. However, we drop an important assumption on the features of the optimal solution made by [28] (see the corresponding Theorem 4.2 and footnote 3), which was only verified experimentally. In a nutshell, we provide the first theoretical validation of the occurrence of DNC in practice: deep neural collapse gives a globally optimal representation of the data under the unconstrained features assumption, for any arbitrary depth.

In addition to our theoretical contribution, we numerically validate in Section 4.2 the occurrence of DNC when training the DUFM with gradient descent. The loss achieved by the DUFM after training agrees well with the theoretically computed optimum, and the corresponding solution clearly exhibits the properties described by our main result. Finally, to check whether the main assumption of the DUFM – the unconstrained features – is relevant in practical training, we show that DNNs trained on CIFAR-10 can indeed represent features suitable for the occurrence of DNC. In particular, *(i)* we pre-train a small DUFM to full collapse, *(ii)* attach the fully-connected layers (DUFM without the unconstrained features) to the end of a randomly initialized CNN, and finally *(iii)* train the CNN with the DUFM's weights fixed. The plots in Figure 1 show that the backbone recovers the unconstrained features leading to DNC. This is quite remarkable, as the collapse, being data-dependent, completely breaks after exchanging the trained unconstrained features with the output of the randomly initialized backbone. Additional details can be found in Section 4.2.

## 2   Related work

Since the original paper [22], neural collapse has been intensively studied, with [21, 3] introducing the now widely accepted unconstrained features model (UFM), also called layer peeled model. In particular, the work on NC can be loosely categorized into *(i)* the study of the mathematical grounds behind NC [31, 19, 40, 3, 42, 12, 21, 7, 18, 29, 36, 37], *(ii)* the development of a connection between NC and DNN performance [32, 5, 6, 1, 34, 15, 39, 17, 16] and *(iii)* empirical observations concerning NC [35, 20]. The first line of work can be further divided into *(i-a)* a static analysis, that studies the global optimality of NC on suitable objectives [31, 19, 40, 28, 42, 12], and *(i-b)* a dynamic analysis, that characterizes the emergence of NC in training with gradient-based methods [21, 7, 12, 30]. The vast majority of works uses the UFM to derive their results, but there are attempts to derive NC without it, see e.g. [25, 33, 23, 24, 14, 38]. For additional references, see also the recent survey [13].

More specifically, [31, 19] show global optimality of NC under UFM, when using the cross-entropy (CE) loss. Similar optimality results are obtained in [40, 28] for the MSE loss. For CE loss, a class-imbalanced setting is considered in [3], where the danger of minority classes collapsing to a single representation (thus being indistinguishable by the network) is pointed out. This analysis is further refined in [27], where a generalization of the ETF geometry under class imbalance is discovered. On the edge between static and dynamic analysis are results which characterize the optimization landscape of UFM. They aim to show that all the stationary points are either global optima (thus exhibiting NC) or local maxima/strict saddle points with indefinite Hessian, which enables optimization methods capable of escaping strict saddles to reach the global optimum. For the CE loss, this property is proved in [42] and [12], while in the MSE setup this insight is provided by [40]. A unified analysis for CE, MSE and some other losses is done in [41]. The convergence of the GD dynamics to NC is considered in [21] under MSE loss for near-zero initialization. The more refined analysis of [7] reveals that the last layer's weights of UFM are close to being conditionally optimal given the features and, assuming their *exact* optimality, the emergence of NC is demonstrated with renormalized gradient flow. For CE loss, [12] shows that gradient flow converges in direction to the KKT point of the max-margin optimization problem, *even without* any explicit regularization (unlike in the MSE loss setting). As the optimal solutions of this max-margin problem exhibit NC, this yields the convergence of gradient-based methods to NC as well. Quantitative results on the speed of convergence for both the MSE and CE losses are given in [30].

Most relevant to our paper, the emergence of NC on earlier layers of DNNs is empirically described in [8, 26, 11, 4]. In particular, it is demonstrated in [8] that, at the end of training, the NC1 metric decreases log-linearly with depth, thus showing a clear propagation of neural collapse. Most recently, the propagation of all the NC properties beyond the last layer is studied in [26], for a setting that includes biases. On the theoretical side, the UFM with *multiple linear* layers is considered in [2], where it is shown that NC is globally optimal for different formulations of the problem. The first result for a non-linear model with *two layers* (separated by a ReLU activation) is given by [28]. There, it is shown that NC, as formulated in the bias-free setting, is globally optimal. However, an additional assumption on the optimum is required, see the detailed discussion in Section 4.

## 3   Preliminaries

**Notation.**   Let $N = Kn$ be the total number of training samples, where $K$ is the number of classes and $n$ the number of samples per class. We are interested in the last $L$ layers of a DNN. The layers are counted from the beginning of indexing, i.e., the $L$-th layer denotes the last one. The layers are fully connected and bias-free with numbers of input neurons $d_1, \ldots, d_L > 1$ and weight matrices $W_1 \in \mathbb{R}^{d_2 \times d_1}, W_2 \in \mathbb{R}^{d_3 \times d_2}, \ldots, W_L \in \mathbb{R}^{K \times d_l}$. Let $H_1 \in \mathbb{R}^{d_1 \times N}, H_2 \in \mathbb{R}^{d_2 \times N}, \ldots, H_L \in \mathbb{R}^{d_L \times N}$ be the feature matrices *before* the application of the ReLU activation function, which is denoted by $\sigma$, i.e., $H_2 = W_1 H_1$ and $H_l = W_{l-1}\sigma(H_{l-1})$ for $l \geq 3$. When indexing a particular sample, we denote by $h_{c,i}^{(l)}$ the $i$-th sample of the $c$-th class in the $l$-th layer, i.e., a particular column of $H_l$. Moreover, let $\mu_c^{(l)} = \frac{1}{n}\sum_{i=1}^{n} h_{c,i}^{(l)}$ be the activation class-mean for layer $l$ and class $c$, and $\bar{H}_l$ the matrix containing $\mu_c^{(l)}$ as the $c$-th column. For convenience, the training samples are arranged class-by-class (first all the samples of first class, then of the second, etc.). Thus, the label matrix containing the one-hot encodings $Y \in \mathbb{R}^{K \times N}$ can be written as $I_K \otimes \mathbf{1}_n^T$, where $I_K$ denotes a $K \times K$ identity matrix, $\otimes$

denotes the Kronecker product and $\mathbf{1}_n$ a row vector of all-ones of length $n$. With this, we can write $\bar{H}_l = \frac{1}{n} H_l (I_K \otimes \mathbf{1}_n)$ and we further denote $\overline{\sigma(H_l)} := \frac{1}{n} \sigma(H_l)(I_K \otimes \mathbf{1}_n)$.

**Deep neural collapse.** Since we assume the last $L$ layers to be bias-free, we formulate a version of the deep neural collapse (DNC) without biases (for a formulation including biases, see [26]).

**Definition 1.** *The deep neural collapse at layer $l$ is described by the following three properties:*[2]

DNC1: *The within-class variability after $l-1$ layers is $0$. This property can be stated for the features either before or after the application of the activation function $\sigma$. In the former case, the condition requires $h_{c,i}^{(l)} = h_{c,j}^{(l)}$ for all $i, j \in [n]$ (or, in matrix notation, $H_l = \bar{H}_l \otimes \mathbf{1}_n^T$); in the latter, $\sigma(h_{c,i}^{(l)}) = \sigma(h_{c,j}^{(l)})$ for all $i, j \in [n]$ (or, in matrix notation, $\sigma(H_l) = \overline{\sigma(H_l)} \otimes \mathbf{1}_n^T$).*

DNC2: *The feature representations of the class-means after $l-1$ layers form an orthogonal matrix. As for DNC1, this property can be stated for features either before or after $\sigma$. In the former case, the condition requires $\bar{H}_l^T \bar{H}_l \propto I_K$; in the latter, $\overline{\sigma(H_l)}^T \overline{\sigma(H_l)} \propto I_K$.*

DNC3: *The rows of the weight matrix $W_l$ are either $0$ or collinear with one of the columns of the class-means matrix $\overline{\sigma(H_l)}$.*

Note that, in practice, DNNs do not achieve such properties *exactly*, but they approach them as the training progresses.

**Deep unconstrained features model.** Next, we generalize the standard unconstrained features model and its extensions [28, 2] to its deep, non-linear counterpart.

**Definition 2.** *The $L$-layer deep unconstrained features model (L-DUFM) denotes the following optimization problem:*

$$\min_{H_1, W_1, \ldots, W_L} \frac{1}{2N} \| W_L \sigma(W_{L-1} \sigma(\ldots W_2 \sigma(W_1 H_1))) - Y \|_F^2 + \sum_{l=1}^{L} \frac{\lambda_{W_l}}{2} \| W_l \|_F^2 + \frac{\lambda_{H_1}}{2} \| H_1 \|_F^2, \tag{1}$$

*where $\|\cdot\|_F$ denotes the Frobenius norm and $\lambda_{H_1}, \lambda_{W_1}, \ldots, \lambda_{W_L} > 0$ are regularization parameters.*

## 4 Optimality of deep neural collapse

We are now state our main result (with proof sketch in Section 4.1, full proof in Appendix A).

**Theorem 3.** *The optimal solutions of the L-DUFM (1) for binary classification ($K = 2$) are s.t.*

$$(H_1^*, W_1^*, \ldots, W_L^*) \quad \text{exhibits DNC}, \qquad \text{if} \quad n \lambda_{H_1} \lambda_{W_1} \ldots \lambda_{W_L} < \frac{(L-1)^{L-1}}{2^{L+1} L^{2L}}. \tag{2}$$

*More precisely, DNC1 is present on all layers; DNC2 is present on $\sigma(H_l^*)$ for $l \geq 2$ and on $H_l^*$ for $l \geq 3$; and DNC3 is present for $l \geq 2$. The optimal $W_1^*, H_1^*$ do not necessarily exhibit DNC2 or DNC3. If the inequality in (2) holds with the opposite strict sign, then the optimal solution is only $(H_1^*, W_1^*, \ldots, W_L^*) = (0, 0, \ldots, 0)$.*

In words, Theorem 3 shows that, unless the regularization is so strong that the optimal solution is all-zeros, the optimal solutions of the DUFM exhibit all the DNC properties. Notice that (2) requires the per-layer regularization to be at most of order $L^{-1} n^{1/(L+1)}$, which is a very mild restriction. If the condition in (2) holds with equality, the set of optimal solutions includes *(i)* solutions exhibiting DNC, *(ii)* the all-zero solution, and *(iii)* additional solutions that do *not* exhibit DNC.

We highlight that Theorem 3 holds for *arbitrarily deep non-linear* unconstrained features models. This is a significant improvement over both [28], which considers $L = 2$, and [2], which considers the linear version of the DUFM. Furthermore, Theorem 4.2 of [28] has a similar statement for two layers

---

[2]NC often involves an additional fourth property – the last layer acting as a nearest class classifier. This property is well defined only for networks including biases, hence we will not consider it here.

(although without considering the DNC1 property on $H_1, H_2$), but it requires the extra assumption $\|\sigma(H_2^*)\|_* = \|H_2^*\|_*$, where $\|\cdot\|_*$ denotes the nuclear norm. In particular, the argument of [28] crucially relies on the fact that $\|\sigma(H_2^*)\|_* \leq \|H_2^*\|_*$. We remark that the inequality $\|\sigma(M)\|_* \leq \|M\|_*$ does not hold for a generic matrix $M$ (see Appendix B for a counterexample) and, at the optimum, it is only observed empirically in [28]. In summary, our result is the first to theoretically address the emerging evidence [26, 8, 4] that neural collapse is not only a story of a single layer, but rather affects multiple layers of a DNN.

## 4.1 Proof sketch for Theorem 3

First, denote by $\mathcal{S}(\cdot)$ an operator which takes any matrix $A \in \mathbb{R}^{e \times f}$ as an input and returns the set of $\min\{e, f\}$ singular values of $A$. The singular values are denoted by $s_i$ and ordered non-increasingly. We specifically label $s_{l,j}$ the $j$-th singular value of $\sigma(H_l)$ for $l \geq 2$.

The argument consists of three steps. First, we assume $n = 1$, and prove that the optimal solutions of the $L$-DUFM problem exhibit DNC2-3 (as stated in Theorem 3). Second, we consider $n > 1$ and show that DNC1 holds. Third, we conclude that DNC2-3 hold for $n > 1$.

**Step 1: DNC2-3 for $n = 1$.** The idea is to split the optimization problem (1) into a series of simpler sub-problems, where in each sub-problem we solve for a fraction of the free variables and condition on all the rest. Crucially, the optimal solutions of all the sub-problems only depend on the singular values of $\sigma(H_l)$, for $l \geq 2$. Thus, the final objective can be reformulated in terms of $\mathcal{S}(\sigma(H_l))$, for $l \geq 2$. By noticing that the objective is symmetric and separable w.r.t. the first and second singular values of the matrices, we can solve separately for $\{s_{l,1}\}_{2 \leq l \leq L}$ and $\{s_{l,2}\}_{2 \leq l \leq L}$. At this point, we show that these separated problems have a unique global optimum, which is either $0$ or element-wise positive (depending on the values of the regularization parameters $\lambda_{H_1}, \lambda_{W_1}, \ldots, \lambda_{W_L}$). Hence, the optimal $s_{l,j}$ will be equal for $j = 1$ and $j = 2$, which in turn yields that $\sigma(H_l)$ is orthogonal. Finally, using the intermediate steps of this procedure, we are able to reverse-deduce the remaining properties.

We now sketch the step of splitting the problem into $L$ sub-problems. As the $l$-th sub-problem, we optimize over matrices with index $L - l + 1$ and condition on all the other matrices, as well as singular values of $\{\sigma(H_l)\}_{l \geq 2}$ (keeping them fixed). We start by optimizing over $W_L$ and then optimize over $W_l$ for $2 \leq l \leq L - 1$ (for all $l$ the optimization problem has the same form and thus this part is compressed into a single lemma). If $L = 2$, we do not have any $l$ for which $2 \leq l \leq L - 1$ and, hence, we skip this part. Finally, we optimize jointly over $W_1, H_1$. We start by formulating the sub-problem for $W_L$.

**Lemma 4.** *Fix $H_1, W_1, \ldots, W_{L-1}$. Then, the optimal solution of the optimization problem*

$$\min_{W_L} \quad \frac{1}{4} \|W_L \sigma(W_{L-1}(\ldots W_2 \sigma(W_1 H_1))) - I_2\|_F^2 + \frac{\lambda_{W_L}}{2} \|W_L\|_F^2$$

*is achieved at $W_L = \sigma(H_L)^T (\sigma(H_L)\sigma(H_L)^T + 2\lambda_{W_L} I_{d_L})^{-1}$ and equals*

$$\frac{\lambda_{W_L}}{2(s_{L,1}^2 + 2\lambda_{W_L})} + \frac{\lambda_{W_L}}{2(s_{L,2}^2 + 2\lambda_{W_L})}. \tag{3}$$

The proof follows from the convexity of the problem in $W_L$ and a simple form of the derivative, see the corresponding lemma in Appendix A for details.

We remark that the solution (3) resulting from the optimization over $W_L$ depends on $\sigma(H_L)$ only through the singular values $s_{L,1}, s_{L,2}$. Thus, we can fix $s_{L,1}, s_{L,2}$ and $H_1, W_1, \ldots, W_{L-2}$, and optimize over $W_{L-1}$ (provided $L - 1 > 1$). It turns out that the resulting optimal value of the objective depends only on $s_{L,1}, s_{L,2}, s_{L-1,1}, s_{L-1,2}$. Thus, we can now fix $s_{L-1,1}, s_{L-1,2}$ and $H_1, W_1, \ldots, W_{L-3}$, and optimize over $W_{L-2}$ (provided $L - 2 > 1$). The optimal solution will, yet again, only depend on the singular values of $\sigma(H_{L-1}), \sigma(H_{L-2})$. Thus, we can fix $s_{L-2,1}, s_{L-2,2}$ and $H_1, W_1, \ldots, W_{L-4}$ and repeat the procedure (provided $L - 3 > 1$). In this way, we optimize over $W_l$ for all $l \geq 2$.

We formulate one step of this nested optimization chain in the following key lemma. Here, $X$ abstractly represents $H_{l-1}$ and $\{s_1, s_2\}$ represent $\{s_{l,1}, s_{l,2}\}$. A proof sketch is provided below and the complete argument is deferred to Appendix A.

**Lemma 5.** *Let $X$ be any fixed two-column entry-wise non-negative matrix with at least two rows, and let its singular values be given by $\{\tilde{s}_1, \tilde{s}_2\} = \mathcal{S}(X)$. Then, for a given pair of singular values $\{s_1, s_2\}$ and a given dimension $d \geq 2$, the optimization problem*

$$\min_W \ \|W\|_F^2, \qquad \text{s.t. } \{s_1, s_2\} = \mathcal{S}(\sigma(WX)),$$

*further constrained so that the number of rows of $WX$ is $d$, has optimal value equal to*

$$\frac{s_1}{\tilde{s}_1} + \frac{s_2}{\tilde{s}_2},$$

*if $X$ is full rank. If $X$ is rank 1, then the optimal value is $\frac{s_1}{\tilde{s}_1}$ as long as $s_2 = 0$, otherwise the problem is not feasible. If $X = 0$, then necessarily $s_1 = s_2 = 0$ with optimal solution 0, otherwise the problem is not feasible.*

*Proof sketch of Lemma 5.* Notice that, for any fixed value of the matrix $A := \sigma(WX)$, there is at least one $W$ which achieves such a value of $A$, unless the rank of $A$ is bigger than the rank of $X$, in which case the problem is not feasible (as mentioned in the statement). Therefore, we can split the optimization into two nested sub-problems. The inner sub-problem fixes an output $A$ for which $\{s_1, s_2\} = \mathcal{S}(A)$ and optimizes only over all $W$ for which $A = \sigma(WX)$. The outer problem then takes the optimal value of this inner problem and optimizes over all the feasible outputs $A$ satisfying the constraint on the singular values. We phrase the nested sub-problems as two separate lemmas.

**Lemma 6.** *Let $A$ with columns $a, b$ and $X$ with columns $x, y$ be fixed two-column matrices with at least two rows, where $X$ is entry-wise non-negative. Then, the optimal value of $\|W\|_F^2$ subject to the constraint $A = \sigma(WX)$ is*

$$\frac{a^T a \cdot y^T y - 2a^T b \cdot x^T y + b^T b \cdot x^T x}{x^T x \cdot y^T y - (x^T y)^2}, \tag{4}$$

*unless $x$ and $y$ are aligned. In that case, the optimal value is $\frac{b^T b}{y^T y}$, and we require that $a, b$ are aligned and $\|y\| / \|x\| = \|b\| / \|a\|$ for the problem to be feasible. Moreover, the matrix $W^* X$ at the optimal $W^*$ is non-negative and thus $W^* X = \sigma(W^* X)$. If the matrix $X$ is orthogonal, then the rows of the optimal $W^*$ are either 0, or aligned with one of the columns of $X$.*

*Proof sketch of Lemma 6.* The optimization problem is separable with respect to the rows of $W$, hence we can solve for each row separately. As the problem is convex and has linear constraints, the KKT conditions are necessary and sufficient for optimality. By solving such conditions explicitly and computing the objective value, the result follows. The details are deferred to Appendix A. □

Now, we can plug (4) into the outer optimization problem, where we solve for the output matrix $A$ given the constraint on singular values. We only optimize over the numerator of (4), as the denominator does not depend on the variable.

**Lemma 7.** *Let $X$ be a fixed entry-wise non-negative rank-2 matrix (the rank-1 case is treated separately later) with columns $x, y$ and $A = \sigma(WX)$. The minimum of the following optimization problem over this non-negative matrix $A$ with columns $a, b$:*

$$\min_A \ b^T b x^T x - 2a^T b x^T y + a^T a y^T y, \qquad \text{s.t. } \mathcal{S}(A) = \{s_1, s_2\} \tag{5}$$

*is $\frac{1}{2}(s_1^2 + s_2^2)(\tilde{s}_1^2 - \tilde{s}_2^2) + \frac{1}{2}(s_1^2 - s_2^2)(\tilde{s}_1^2 - \tilde{s}_2^2)$, where $\{\tilde{s}_1, \tilde{s}_2\} = \mathcal{S}(X)$.*

*Proof sketch of Lemma 7.* The optimization problem (5) is non-convex and has non-linear constraints. It is still solvable using the KKT conditions, however some care is required as such conditions are no longer necessary and sufficient. Using the Linear Independence Constraint Qualification (LICQ) rule, we handle the necessity. Then, by solving for KKT conditions and comparing the objective value with solutions for which the LICQ condition does not hold, we obtain the desired description of the optimal solution. The details are deferred to Appendix A. □

At this point, we are ready to obtain the result of Lemma 5. If $\tilde{s}_1, \tilde{s}_2 \neq 0$ then, by combining the result of Lemma 7 with (4), the proof is complete. If $\tilde{s}_2 = 0$, we can optimize (4), given $s_2 = 0$ and the extra constraint $\frac{a^T a}{b^T b} = \frac{x^T x}{y^T y}$, which follows from the requirement on the rank of $A$ and the fact that $X$ has rank 1. The claim then follows from a direct computation of the objective value using the constraint. The case $\tilde{s}_1 = \tilde{s}_2 = 0$ gives that $X = 0$, and is trivial. □

So far, we have reduced the optimization over $\{W_l\}_{l=2}^{L-1}$ to the optimization over the singular values of $\{\sigma(H_l)\}_{l=2}^{L}$. We now do the same with the first layer's matrices $W_1, H_1$. For this, we again split the optimization of $\frac{\lambda_{W_1}}{2}\|W_1\|_F^2 + \frac{\lambda_{H_1}}{2}\|H_1\|_F^2$ given $\{s_1, s_2\} = \mathcal{S}(\sigma(W_1 H_1))$ into two nested sub-problems: the inner one performs the optimization for a fixed output $H_2 := W_1 H_1$; the outer one optimizes over the output $H_2$, given the optimal value of the inner problem.

The inner sub-problem is solved by the following Lemma, which describes a variational form of the nuclear norm. The statement is equivalent to Lemma C.1 of [28].

**Lemma 8.** *The optimization problem*

$$\min_{A,B;C=AB} \quad \frac{\lambda_A}{2}\|A\|_F^2 + \frac{\lambda_B}{2}\|B\|_F^2 \tag{6}$$

*is minimized at value $\sqrt{\lambda_A \lambda_B}\|C\|_*$, and the minimizers are of the form $A^* = \gamma_A U\Sigma^{1/2}R^T$, $B^* = \gamma_B R\Sigma^{1/2}V^T$. Here, the constants $\gamma_A, \gamma_B$ only depend on $\lambda_A, \lambda_B$; $U\Sigma V^T$ is the SVD of $C$; and $R$ is an orthogonal matrix.*

To be more concrete, we apply Lemma 8 with $A = W_1$ and $B = H_1$ (so that $C = W_1 H_1 = H_2$). Thus, we are only left with optimizing $\|H_2\|_*$ given $\mathcal{S}(\sigma(H_2)) = \{s_1, s_2\}$. We solve this problem in the following, much more general lemma, which might be of independent interest. Its proof is deferred to Appendix A.

**Lemma 9.** *Let $L \geq 2$ be a positive integer. Then, the optimal value of the optimization problem*

$$\min_H \|H\|_{S_{\frac{2}{L}}}^{\frac{2}{L}}, \qquad \text{s.t.} \quad \mathcal{S}(\sigma(H)) = \{s_1, s_2\} \tag{7}$$

*equals $(s_1 + s_2)^{\frac{2}{L}}$. Here $\|\cdot\|_{S_p}$ denotes the $p$-Schatten pseudo-norm.*

**Remark 10.** *As a by-product of the analysis in the proof of Lemma 9, we obtain a characterization of the minimizers of (7), which will be useful in **Step 2**. Let $H^*$ be an optimizer of (7). Then, $\sigma(H^*)$ has orthogonal columns. For $L > 2$, its negative entries are set uniquely so that $H^*$ has rank 1. For $L = 2$, its negative entries are arranged as follows. All the rows of $\sigma(H^*)$ which contain exactly one positive entry on the first column will result in rows of $H^*$ that are multiples of each other. Similarly, all the rows of $\sigma(H^*)$ which contain exactly one positive entry on the second column will result in rows of $H^*$ that are multiples of each other. The zero rows of $\sigma(H^*)$ remain zero rows in $H^*$. Moreover, the $l_2$ norm of the negative entries of the rows of $H^*$ with positive entry in the first column equals the $l_2$ norm of the negative entries of the rows of $H^*$ where the positive entry is in the second column. Finally, taking any two rows of $H^*$ with counterfactual positioning of the signs of the entries, the product of negative entries does not exceed the product of positive entries.*

Armed with Lemma 4, 5 and 9, we reformulate the $L$-DUFM objective (1) only in terms of $\{s_{l,1}, s_{l,2}\}_{2\leq l\leq L}$. We then split this joint problem into two separate, identical optimizations over $\{s_{l,1}\}_{2\leq l\leq L}$ and $\{s_{l,2}\}_{2\leq l\leq L}$, which are solved via the lemma below (taking $x_l := s_{l,1}^2$ and $x_l := s_{l,2}^2$, respectively). Its proof is deferred to Appendix A.

**Lemma 11.** *The optimization problem*

$$\min_{x_l;2\leq l\leq L} \quad \frac{\lambda_{W_L}}{2(x_L + 2\lambda_{W_L})} + \sum_{l=2}^{L-1}\left(\frac{\lambda_{W_l}}{2}\frac{x_{l+1}}{x_l}\right) + \sqrt{\lambda_{W_1}\lambda_{H_1}}\sqrt{x_2} \tag{8}$$

*is optimized at an entry-wise positive, unique solution if*

$$\lambda_{H_1}\prod_{l=1}^{L}\lambda_{W_l} < \frac{(L-1)^{L-1}}{2^{L+1}L^{2L}}, \tag{9}$$

*and at $(x_2, \ldots, x_L) = (0, \ldots, 0)$ otherwise. When (9) holds with equality, both solutions are optimal. Here, $0/0$ is defined to be 0.*

Note that (9) is equivalent to (2) when $n = 1$. Furthermore, if the optimization problem (8) has a single non-zero solution, then the optimal $s_{l,1}^*$ and $s_{l,2}^*$ are forced to coincide for all $l$. From this, we obtain the orthogonality of $\sigma(H_l^*)$ for $l \geq 2$. Lemma 6 then gives that the optimal $H_l^*$ for $l \geq 3$ must be orthogonal as well, since it is non-negative and equal to $\sigma(H_l^*)$. Finally, the statement about $W_l^*$ for $l \geq 2$ also follows from Lemma 6. This concludes the proof of DNC2-3 for $n = 1$. As a by-product of Lemma 8, we additionally obtain a precise characterization of the optimal $W_1^*, H_1^*$.

**Step 2: DNC1 for $n > 1$.** Now, we assume $n > 1$ and the conclusion of **Step 1**. We show that DNC1 holds for the optimizer of (1). We give a sketch here, deferring the proof to Appendix A.

We proceed by contradiction. Assume there exist $\tilde{H}_1^*, W_1^*, \ldots, W_L^*$ which achieve the minimum of (1), but $\tilde{H}_1^*$ does *not* exhibit DNC1. Note that the objective in (1) is separable w.r.t. the columns of $H_1$. Thus, for both classes $c \in \{1, 2\}$, the partial objectives corresponding to the columns $\{h_{c,i}^{(1)}\}_{i=1}^n$ of $\tilde{H}_1^*$ must be equal for all $i$. Indeed, if that is not the case, we can exchange all the columns of $\tilde{H}_1^*$ with the ones that achieve the smallest partial objective value and obtain a contradiction. Now, we can construct an alternative $\bar{H}_1^*$ such that, for each class $c$, we pick any $h_{c,i}^{(1)}$ and place it instead of $\{h_{c,j}^{(1)}\}_{j \neq i}$. By construction, $\bar{H}_1^*$ exhibits DNC1 and it is still an optimal solution. At this point, let us construct $H_1^*$ from $\bar{H}_1^*$ by taking a single sample from both classes. We claim that $(H_1^*, W_1^*, \ldots, W_L^*)$ is an optimal solution of (1) with $n = 1$ and $n\lambda_{H_1}$ as the regularization term for $H_1$, while keeping $d_1, \ldots, d_L$ and $\lambda_{W_1}, \ldots, \lambda_{W_L}$ the same. To see why this is the case, consider an alternative $G_1^*$ achieving a smaller loss. Then, $(G_1^* \otimes \mathbf{1}_n^T, W_1^*, \ldots, W_L^*)$ would achieve a smaller loss than $(\bar{H}_1^*, W_1^*, \ldots, W_L^*)$, which contradicts its optimality.

Recall that $H_1^*$ is an optimal solution of (1) with $n = 1$. By **Step 1**, the columns $x, y$ of $\sigma(H_2^*)$ are such that $x, y \geq 0$, $x^T y = 0$, and $\|x\| = \|y\|$ (here, by the notation $x \geq 0$ we mean an entry-wise inequality). Assume $\sigma(H_2^*) \neq 0$. First we show that $x, y$ and thus the whole $\sigma(H_2^*)$ are uniquely determined by $(W_1^*, W_2^*, \ldots, W_L^*)$. For this we use the uniqueness of the scaling obtained in Lemma 11 and the DNC3 property for the $(\sigma(H_2^*), W_2^*)$ pair. Then, by Lemma 9, all the matrices $A_2$ such that $\sigma(A_2) = \sigma(H_2^*)$ and $\|A_2\|_* = \|H_2^*\|_*$ have columns of the form $x - ty, y - tx$ for $t \in [0, 1]$. Let $A_2^t$ be a matrix of that form for a particular $t$. A series of steps which include computing the SVD of $A_2^t$ and the usage of Lemma 8 reveal that, among $0 \leq t \leq 1$, there is a single $t^*$ and $A_2^* = A_2^{t^*}$ such that $W_1^*$ solves the optimization problem (6) in Lemma 8 if $H_2 = A_2^t$. Thus, having fixed $A_2^*$ that can be the output of the first layer, this in turn gives the uniqueness of the unconstrained features $H_1$ solving (6) while $W_1 = W_1^*, H_2 = H_2^* = A_2^*$. However, as $\tilde{H}_1^*$ is not DNC1 collapsed by assumption, there are multiple possible choices for $\bar{H}_1^*$ and then $H_1^*$, which gives a contradiction with the uniqueness of $H_1$ solving (6).

**Step 3: DNC2-3 for $n > 1$.** Having established DNC1, we obtain DNC2-3 by reducing again to the case $n = 1$, which allows us to conclude the proof of the main result. We finally remark that $W_1^*$ and $H_1^*$ do not need to be orthogonal, due to the $\Sigma^{1/2}$ multiplication in the statement of Lemma 8. This means that $W_1^*$ and $H_1^*$ may not exhibit DNC2-3 (as stated in Theorem 3).

### 4.2 Numerical results

**DUFM training.** To support the validity of our theoretical result and to demonstrate that the global optimum in the DUFM can be found efficiently, we conduct numerical experiments in which we solve (1) with randomly initialized weights using gradient descent. In Figure 2, we plot the training progression of the DNC metrics for two choices of the depth: $L = 3$ (first row) and $L = 6$ (second row). In both cases, we use $n = 50$ training samples and constant layer width of $64$. The learning rate is $0.5$, though smaller learning rates produce the same results. We run full gradient descent for $10^5$ steps, and the weight decay is set to $5 \cdot 10^{-4}$ for all the layers. The plots are based on 10 experiments and the error bands are computed as one empirical standard deviation to both sides. To measure the DNC occurrence, we plot the following metrics as a function of training. For DNC1, we define $\Sigma_W^l = \frac{1}{N} \sum_{c=1}^2 \sum_{i=1}^n (h_{c,i}^{(l)} - \mu_c^{(l)})(h_{c,i}^{(l)} - \mu_c^{(l)})^T$ as the *within-class* variability at layer $l$ and $\Sigma_B^l = \frac{1}{2} \sum_{c=1}^2 (\mu_c^{(l)} - \mu_G^{(l)})(\mu_c^{(l)} - \mu_G^{(l)})^T$ as the *between-class* variability, where $\mu_G^{(l)}$ is the global feature mean at layer $l$. Then, the DNC1 metric at layer $l$ is $\|\Sigma_W \Sigma_B^\dagger\|_F^2$, with $\dagger$ being the pseudo-inverse operator. This is an upper bound on the trace of the same matrix, which is sometimes used in the literature for measuring NC1. An ideal DNC1 would show 0 on this metric. For DNC2, our metric at layer $l$ is given by $\frac{s_{l,1}}{s_{l,2}}$ (recall that these are the singular values of $\sigma(H_l)$), and we consider the feature matrices both after and before the ReLU activation. An ideal DNC2 would show 1 on this metric. For DNC3, we consider the weighted average sine of the angles between each row of $W_l$ and the closest column of $\sigma(H_l)$; the weights are the $l_2$ norms of the rows of $W_l$. To avoid

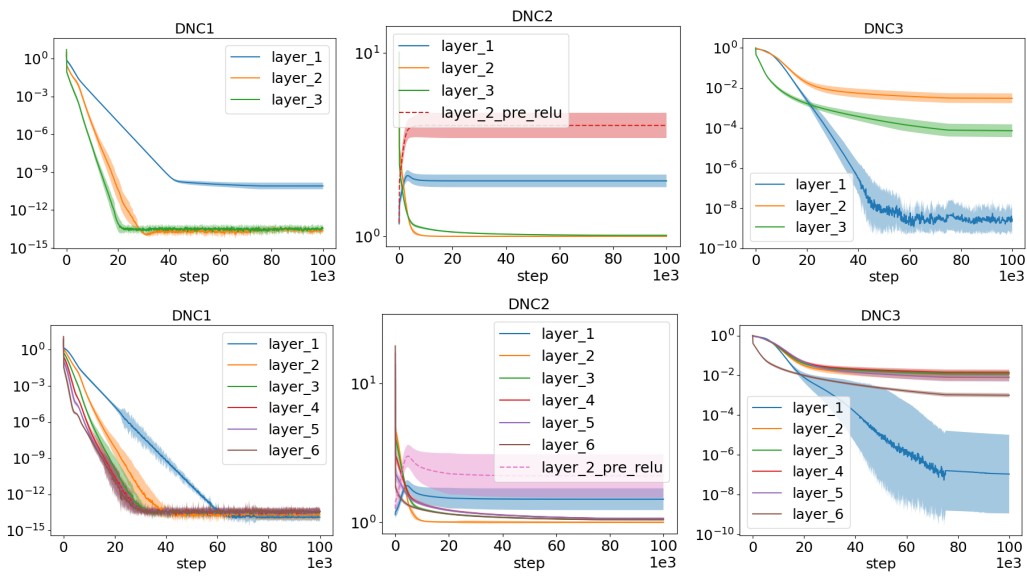

Figure 2: Deep neural collapse metrics as a function of training for the DUFM objective (2). The first (second) row corresponds to $L = 3$ ($L = 6$). The metrics corresponding to DNC1 and DNC3 are very small, and the metric corresponding to DNC2 is close to 1, which means that the solution found by gradient descent exhibits neural collapse in all layers, as predicted by Theorem 3.

numerical issues, we exclude rows of $W_l$ whose $l_2$ norm is smaller than $10^{-6}$. An ideal DNC3 would show 0 on this metric.

The plots in Figure 2 closely follow the predictions of Theorem 3 and, taken as a whole, show that DUFM training via gradient descent clearly exhibits deep neural collapse, for all the three metrics considered, and for both depths. More specifically, DNC1 is progressively achieved in all layers, with the layers closer to the output being faster. DNC2 is achieved for all $\sigma(H_l)$ with $l \geq 2$ and $H_l$ for $l \geq 3$ (though we didn't include this in plots to avoid too many lines) and, as predicted by our theory, it is not achieved on $H_2$ (the dotted line in the middle plot) and on $H_1$ (the blue line). Interestingly, for these numbers of layers, SGD does not find solutions which would exhibit DNC2 on $H_1$, even though such solutions exist. Hence, gradient descent is not implicitly biased towards them, see Appendix C for a more detailed discussion. DNC3 is achieved on all layers – even on the first, which is not covered by our theory and where the effect is the most pronounced; the deeper layers achieve values of DNC3 of order $10^{-2}$, which translates into an angle of less than one degree between weight rows and feature columns. Finally, we highlight that the training loss matches the optimal value computed numerically via (17) (the corresponding plot is reported in Appendix C), and hence the solution found by gradient descent is globally optimal. Additional ablation studies concerning the number of layers, weight decay and width of the layers are reported in Appendix C.

**ResNet20 training with pre-trained DUFM.** We support the validity of DUFM as a modeling principle for deep neural collapse via the following experiments. We first train an $L$-DUFM for $L = 3, 5$ with $K = 2$ and $n = 5000$ to full optimality, as in the previous paragraph. Then, we replace $H_1^*$ with the output of a randomly initialized backbone, we fix the weight matrices in the DUFM, and we train end-to-end to convergence on the classes 0 and 1 of CIFAR-10. In Figure 1, we report the DNC metrics of our 3-DUFM head, as the training of the backbone ResNet20 progresses. While at the start of the training the features $H_1$ are random and, therefore, the model does not exhibit DNC, the ResNet20 recovers well the DNC properties after training. This suggests that the crucial assumption of DUFM – the unconstrained features, indeed holds in practice. The plots are based on 7 experiments with confidence bands of one empirical standard deviation to both sides. Additional details as well as the experiments with 5-DUFM and VGG13 and DenseNet121 are discussed in Appendix D.

## 5  Conclusion

In this work, we propose the deep unconstrained features model (DUFM) – a multi-layered version of the popular unconstrained features model – as a tool to theoretically understand the phenomenon of deep neural collapse (DNC). For binary classification, we show that DNC is globally optimal in the DUFM. This provides the first rigorous evidence of neural collapse occurring for arbitrarily many layers. Our numerical results show that gradient descent efficiently finds a solution in agreement with our theory – one which displays neural collapse in multiple layers. Finally, we provide evidence in favor of DUFM as a modeling principle, by showing that standard DNNs can learn the unconstrained features that exhibit all DNC properties.

Interesting open problems include *(i)* the generalization to multiple classes ($K > 2$), *(ii)* the analysis of the gradient dynamics in the DUFM, and *(iii)* understanding the impact of biases on the collapsed solution, as well as that of the cross-entropy loss. As for the first point, a crucial issue is to control $\|W_1 H_1\|_*$, since the inequality $\|\sigma(M)\|_* \leq \|M\|_*$ does not hold for general matrices $M$. In addition, while Lemma 5 still appears to be valid for $K > 2$ (and it would be an interesting stand-alone result), its extension likely requires a new argument.

## Acknowledgments and Disclosure of Funding

M. M. is partially supported by the 2019 Lopez-Loreta Prize. The authors would like to sincerely thank Eugenia Iofinova, Bernd Prach and Simone Bombari for valuable feedback to our manuscript.

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

# A  Proofs

**Theorem 3.** *The optimal solutions of the L-DUFM* (1) *for binary classification* $(K = 2)$ *are s.t.*

$$(H_1^*, W_1^*, \ldots, W_L^*) \quad \textit{exhibits DNC}, \qquad \textit{if} \quad n\lambda_{H_1}\lambda_{W_1}\ldots\lambda_{W_L} < \frac{(L-1)^{L-1}}{2^{L+1}L^{2L}}. \qquad (2)$$

*More precisely, DNC1 is present on all layers; DNC2 is present on* $\sigma(H_l^*)$ *for* $l \geq 2$ *and on* $H_l^*$ *for* $l \geq 3$; *and DNC3 is present for* $l \geq 2$. *The optimal* $W_1^*, H_1^*$ *do not necessarily exhibit DNC2 or DNC3. If the inequality in* (2) *holds with the opposite strict sign, then the optimal solution is only* $(H_1^*, W_1^*, \ldots, W_L^*) = (0, 0, \ldots, 0)$.

*Proof.* We will split the proof into two distinct cases: $n = 1$ and $n > 1$. The proof for $n > 1$ will require the statement to hold for $n = 1$, so we start with this case. The explicit form of the problem goes as:

$$\min_{H_1, W_1, \ldots, W_L} \frac{1}{2N} \|W_L\sigma(W_{L-1}(\ldots W_2\sigma(W_1 H_1)) - I_2\|_F^2 + \sum_{l=1}^{L} \frac{\lambda_{W_l}}{2} \|W_l\|_F^2 + \frac{\lambda_{H_1}}{2} \|H_1\|_F^2. \quad (10)$$

We will solve this problem using a sequence of $L$ sub-problems, where in sub-problem $l$, we only optimize over matrices with index $L - l + 1$ and condition on all the other matrices, as well as singular values of all $\sigma(H_l)$ for $l \geq 2$ (keeping them fixed). Throughout the process, we will strongly rely on the crucial phenomenon that the optimal solutions of all the sub-problems will only depend on the singular values of the matrices $\sigma(H_l)$. We will solve the problem starting from the last weight matrix $W_L$, through $W_{L-1}$ and so on, and finishing with the joint optimization over $W_1, H_1$. We divide this procedure into 4 lemmas. The first lemma will only treat $W_L$. The second lemma will treat $W_l$ for $2 \leq l \leq L - 1$. This lemma will only be used if $L \geq 3$. The third lemma will treat $W_1, H_1$ jointly and the fourth, final lemma will then treat the only free variables left – the singular values of $\sigma(H_l)$ for all $l \geq 2$. Let us denote by $\mathcal{S}(\cdot)$ an operator which returns a set of singular values of any given matrix (in full SVD, i.e. the SVD with number of singular values equal to the smaller dimension of the matrix). Moreover, let $s_{l,i}$ be the $i$-th singular value of $\sigma(H_l)$.

**Lemma 4.** *Fix* $H_1, W_1, \ldots, W_{L-1}$. *Then, the optimal solution of the optimization problem*

$$\min_{W_L} \quad \frac{1}{4} \|W_L\sigma(W_{L-1}(\ldots W_2\sigma(W_1 H_1)) - I_2\|_F^2 + \frac{\lambda_{W_L}}{2} \|W_L\|_F^2$$

*is achieved at* $W_L = \sigma(H_L)^T(\sigma(H_L)\sigma(H_L)^T + 2\lambda_{W_L} I_{d_L})^{-1}$ *and equals*

$$\frac{\lambda_{W_L}}{2(s_{L,1}^2 + 2\lambda_{W_L})} + \frac{\lambda_{W_L}}{2(s_{L,2}^2 + 2\lambda_{W_L})}. \qquad (3)$$

*Proof.* This is a well-known fact from the theory of ridge regression [10]. However, for readers' convenience, we do the proof here too. If we denote the objective of this as $\mathcal{L}$, then we get:

$$\frac{\partial \mathcal{L}}{\partial W_L} = \frac{1}{2}(W_L\sigma(H_L) - I_2)\sigma(H_L)^T + \lambda_{W_L}W_L.$$

Since the objective is strongly convex in $W_L$, the unique global optimum is found by setting to 0 the derivative. Solving for it, we have

$$W_L = \sigma(H_L)^T(\sigma(H_L)\sigma(H_L)^T + 2\lambda_{W_L} I_{d_L})^{-1}.$$

Let us denote $\sigma(H_L) = U\Sigma V^T$ the rectangular SVD of $\sigma(H_L)$ where $U \in \mathbb{R}^{d_L \times d_L}, \Sigma \in \mathbb{R}^{d_L \times 2}, V \in \mathbb{R}^{2 \times 2}$. Writing the optimal $W_L$ in terms of SVD of $\sigma(H_L)$ we get:

$$\begin{aligned} W_L &= V\Sigma^T U^T(U\Sigma\Sigma^T U^T + 2\lambda_{W_L} I_{d_L})^{-1} \\ &= V\Sigma^T U^T U(\Sigma\Sigma^T + 2\lambda_{W_L} I_{d_L})^{-1}U^T \\ &= V\Sigma^T(\Sigma\Sigma^T + 2\lambda_{W_L} I_{d_L})^{-1}U^T. \end{aligned}$$

Therefore, we obtain the SVD of $W_L$ and that $(\Sigma\Sigma^T + 2\lambda_{W_L} I_{d_L})^{-1}$ is a matrix of singular values of $W_L$, from which we obtain:

$$\|W_L\|_F^2 = \sum_{i=1}^{2} \frac{s_{L,i}^2}{(s_{L,i}^2 + 2\lambda_{W_L})^2}.$$

Furthermore, the first summand of the objective is

$$\|W_L \sigma(H_L) - I_2\|_F^2 = \left\|V\Sigma^T(\Sigma\Sigma^T + 2\lambda_{W_L} I_{d_L})^{-1} U^T U\Sigma V^T - I_2\right\|_F^2$$

$$= \left\|V(\Sigma^T(\Sigma\Sigma^T + 2\lambda_{W_L} I_{d_L})^{-1} U^T U\Sigma)V^T - I_2\right\|_F^2$$

$$= \sum_{i=1}^{2} \left(\frac{s_{L,i}^2}{s_{L,i}^2 + 2\lambda_{W_L}} - 1\right)^2.$$

Putting everything together, we get the following optimal value of the objective:

$$\frac{1}{4}\sum_{i=1}^{2} \left(\frac{s_{L,i}^2}{s_{L,i}^2 + 2\lambda_{W_L}} - 1\right)^2 + \frac{\lambda_{W_L}}{2}\sum_{i=1}^{2}\frac{s_{L,i}^2}{(s_{L,i}^2 + 2\lambda_{W_L})^2}$$

$$= \sum_{i=1}^{2}\frac{2\lambda_{W_L}^2}{2(s_{L,i}^2 + 2\lambda_{W_L})^2} + \frac{\lambda_{W_L}\sigma_1^2}{2(s_{L,i}^2 + 2\lambda_{W_L})^2}$$

$$= \sum_{i=1}^{2}\frac{\lambda_{W_L}}{2(s_{L,i}^2 + 2\lambda_{W_L})},$$

which gives the desired result. $\qquad\square$

Note that the term we optimized for is the only part of the objective in (10) which depends on $W_L$. Now we are ready to state the crucial lemma (necessary when $L \geq 3$), which is presented in an abstract way so it can be easily applied outside of the scope of this work.

**Lemma 5.** *Let $X$ be any fixed two-column entry-wise non-negative matrix with at least two rows, and let its singular values be given by $\{\tilde{s}_1, \tilde{s}_2\} = \mathcal{S}(X)$. Then, for a given pair of singular values $\{s_1, s_2\}$ and a given dimension $d \geq 2$, the optimization problem*

$$\min_{W} \ \|W\|_F^2, \qquad \text{s.t. } \{s_1, s_2\} = \mathcal{S}(\sigma(WX)),$$

*further constrained so that the number of rows of $WX$ is $d$, has optimal value equal to*

$$\frac{s_1}{\tilde{s}_1} + \frac{s_2}{\tilde{s}_2},$$

*if $X$ is full rank. If $X$ is rank 1, then the optimal value is $\frac{s_1}{\tilde{s}_1}$ as long as $s_2 = 0$, otherwise the problem is not feasible. If $X = 0$, then necessarily $s_1 = s_2 = 0$ with optimal solution 0, otherwise the problem is not feasible.*

*Proof.* Notice that, for any fixed value of the matrix $A := \sigma(WX)$, there is at least one $W$ which achieves such a value of $A$, unless the rank of $A$ is bigger than the rank of $X$, however this is a special case explicitly mentioned in the statement of the lemma for which the problem is non-feasible. Therefore, we can split the optimization into solving two nested sub-problems. The inner sub-problem fixes an output $A$ for which $\{s_1, s_2\} = \mathcal{S}(A)$ and optimizes only over all $W$ for which $A = \sigma(WX)$. The outer problem then takes the optimal value of this inner problem and optimizes over all the feasible outputs $A$ satisfying the constraint on the singular values. We will phrase both of the nested sub-problems as separate lemmas:

**Lemma 6.** *Let $A$ with columns $a, b$ and $X$ with columns $x, y$ be fixed two-column matrices with at least two rows, where $X$ is entry-wise non-negative. Then, the optimal value of $\|W\|_F^2$ subject to the constraint $A = \sigma(WX)$ is*

$$\frac{a^T a \cdot y^T y - 2a^T b \cdot x^T y + b^T b \cdot x^T x}{x^T x \cdot y^T y - (x^T y)^2}, \tag{4}$$

*unless $x$ and $y$ are aligned. In that case, the optimal value is $\frac{b^T b}{y^T y}$, and we require that $a, b$ are aligned and $\|y\| / \|x\| = \|b\| / \|a\|$ for the problem to be feasible. Moreover, the matrix $W^* X$ at the optimal $W^*$ is non-negative and thus $W^* X = \sigma(W^* X)$. If the matrix $X$ is orthogonal, then the rows of the optimal $W^*$ are either 0, or aligned with one of the columns of $X$.*

*Proof.* Note that this optimization problem is separable in rows of $W$. Thus it suffices to solve the problem for a single row $w_j$ and then sum up over all rows. We will, moreover, split the analysis into two cases.

**Case 1:** Here we will assume that the row of $A$ we are optimizing for is $(a_j, b_j)$, where $a_j, b_j > 0$. In this case, we are facing the following (simple) convex optimization problem:

$$\min_{w_j} \|w_j\|_2^2$$
$$\text{s.t. } w_j^T x = a_j$$
$$w_j^T y = b_j$$

This has a known solution – the pseudoinverse as it is the minimum interpolating solution of a two-datapoint linear regression. However, using the formula would not simplify the expressions in the slightest and so we will solve this problem manually. The Lagrange function for this problem is

$$\mathcal{L}(w_j, u, v) = \sum_{i=1}^{d} w_{ji}^2 + u \left( \sum_{i=1}^{d} w_{ji} x_i - a_j \right) + v \left( \sum_{i=1}^{d} w_{ji} y_i - b_j \right),$$

where $d$ is the dimension of the columns of $X$ and rows of $W$. We know that solving the KKT conditions for a convex problem with linear constraints yields sufficient and necessary conditions for the optimality. Let us write:

$$\frac{\partial \mathcal{L}}{\partial w_{ji}} = 2 w_{ji} + u x_i + v y_i \stackrel{!}{=} 0.$$

Together with constraints, these are the only KKT conditions for this problem. Solving for $w_{ji}$ we get:

$$w_{ji} = \frac{-u x_i - v y_i}{2}. \tag{11}$$

Plugging in the constraints we get:

$$\sum_{i=1}^{d} v y_i^2 + u x_i y_i = v y^T y + u x^T y = -2 b_j,$$

$$\sum_{i=1}^{d} u x_i^2 + v x_i y_i = u x^T x + v x^T y = -2 a_j.$$

We can solve for $v$ from the first equation and plug in to the second equation. Since $y^T y$ must be non-zero, otherwise the constraint $w_j^T y = b_j > 0$ could not be satsfied, we know $v$ and $u$ are well-defined. We get:

$$u = \frac{2 b_j x^T y - 2 a_j y^T y}{x^T x y^T y - (x^T y)^2}, \tag{12}$$

unless $x = \alpha y$, which we will deal with later. Now we can plug $u$ in the expression on $v$ and after some computations we get

$$v = \frac{2 a_j x^T y - 2 b_j x^T x}{x^T x y^T y - (x^T y)^2}.$$

Combining, we get the optimal $w_j$ :

$$\frac{b_j x^T x - a_j x^T y}{x^T x y^T y - (x^T y)^2} y + \frac{a_j y^T y - b_j x^T y}{x^T x y^T y - (x^T y)^2} x.$$

If $x = \alpha y$, then it is clear from (12) that $b_j x^T y = a_j y^T y$, from which we can deduce $\alpha = a_j / b_j$. From the expression (11) for $w_{ji}$, it is clear that the only solution in this case is $w_j = \frac{b_j}{y^T y} y$ and the same must hold for all other $j \in \{1, \ldots, \tilde{d}\}$, where $\tilde{d}$ is the number of rows of $W$.

**Case 2:** Here we will assume that the row of $A$ we are optimizing for is, w.l.o.g. $(a_j, 0)$, where $a_j > 0$. In this case, we are facing the following (simple) convex optimization problem:

$$\min_{w_j} \|w_j\|_2^2$$
$$\text{s.t. } w_j^T x = a_j$$
$$w_j^T y \leq 0$$

Again, we can write the Lagrange function for this problem and again, solving for KKT conditions will provide full description of the optimal solution:

$$\mathcal{L}(w_j, u, v) = \sum_{i=1}^{d} w_{ji}^2 + u \sum_{i=1}^{d} w_{ji} y_i + v \left( \sum_{i=1}^{d} w_{ji} x_i - a_j \right).$$

We obtain the same condition for $w_{ji}$ as in case 1. However, since one of our constraints is now an inequality, we will deal with complementarity conditions. Let us first assume that $u = 0$. In that case $w_{ji} = -v x_i / 2$ and from the first constraint

$$v = \frac{-2a_j}{\sum x_i^2}.$$

Thus, $w_j = \frac{a_j}{x^T x} x$. However judging from this, to satisfy the first constraint, we must have $x_i y_i = 0$ for all $i$ (remember that $\sum w_{ji} y_i$ must be non-positive and that $x, y$ have non-negative entries). Therefore necessarily $x^T y = 0$. However, looking at the solution of case 1, we see that after plugging in $x^T y = 0$ and $b_j = 0$, we get the same expression. Thus, this is a special case of case 1. The second option for complementarity condition is that $\sum w_{ji} y_i = 0$. Then, we can proceed in the same way as in case 1 to obtain the same formula after plugging in $b = 0$. Then again, this is a special case of the formula in case 1.

All in all, we have fully solved the optimization problem in question. Now it remains to compute the optimal value. In the special case that $x = \alpha y$, we must have $b x^T y = a y^T y$, otherwise there is no feasible solution. In this case this is equivalent to $\alpha = a/b, w_j = b_j/(y^T y) y$ and $w_j^T w_j = b_j^2/(y^T y)$. Therefore, summing over all rows of $W$ (and keeping in mind that once $x = \alpha y$, the same characterization holds for every row) we get $\|W\|_F^2 = \frac{b^T b}{y^T y}$.

In the general case, let us compute:

$$w_j^T w_j = \left\langle \frac{b_j x^T x - a_j x^T y}{x^T x y^T y - (x^T y)^2} y + \frac{a_j y^T y - b_j x^T y}{x^T x y^T y - (x^T y)^2} x, \frac{b_j x^T x - a_j x^T y}{x^T x y^T y - (x^T y)^2} y + \frac{a_j y^T y - b_j x^T y}{x^T x y^T y - (x^T y)^2} x \right\rangle$$

$$= \frac{(b_j x^T x - a_j x^T y)^2 y^T y + 2(b_j x^T x - a_j x^T y)(a_j y^T y - b_j x^T y) x^T y + (a_j y^T y - b_j x^T y)^2 x^T x}{(x^T x y^T y - (x^T y)^2)^2}$$

$$= \frac{b_j^2 (x^T x)^2 y^T y - 2 b_j a_j x^T x y^T y x^T y + a_j^2 (x^T y)^2 y^T y + 2 b_j a_j x^T x y^T y x^T y - 2 b_j^2 x^T x (x^T y)^2}{(x^T x y^T y - (x^T y)^2)^2}$$

$$+ \frac{-2 a_j^2 y^T y (x^T y)^2 + 2 b_j a_j (x^T y)^3 + a_j^2 (y^T y)^2 x^T x - 2 b_j a_j x^T x y^T y x^T y + b_j^2 (x^T y)^2 x^T x}{(x^T x y^T y - (x^T y)^2)^2}$$

$$= \frac{b_j^2 (x^T x)^2 y^T y - b_j^2 x^T x (x^T y)^2 + a_j^2 (y^T y)^2 x^T x - a_j^2 y^T y (x^T y)^2 - 2 a_j b_j x^T x y^T y + 2 a_j b_j (x^T y)^3}{(x^T x y^T y - (x^T y)^2)^2}$$

$$= \frac{(b_j^2 x^T x + a_j^2 y^T y - 2 a_j b_j x^T y)(x^T x y^T y - (x^T y)^2)}{(x^T x y^T y - (x^T y)^2)^2}$$

$$= \frac{b_j^2 x^T x - 2 a_j b_j x^T y + a_j^2 y^T y}{x^T x y^T y - (x^T y)^2}.$$

Summing up over rows of $W$, we get the desired expression:

$$\|W\|_F^2 = \frac{b^T b x^T x - 2 a^T b x^T y + a^T a y^T y}{x^T x y^T y - (x^T y)^2}.$$

The rest of the statement of the lemma clearly follows from the formula on optimal rows of $W$. $\qquad\square$

Now, as promised, we will optimize over the output matrix $A$ given that we know what the partial optimal value for each $A$ is.

**Lemma 7.** *Let $X$ be a fixed entry-wise non-negative rank-2 matrix (the rank-1 case is treated separately later) with columns $x, y$ and $A = \sigma(WX)$. The minimum of the following optimization problem over this non-negative matrix $A$ with columns $a, b$:*

$$\min_A \; b^T b x^T x - 2a^T b x^T y + a^T a y^T y, \qquad s.t. \; \mathcal{S}(A) = \{s_1, s_2\} \qquad (5)$$

*is $\frac{1}{2}(s_1^2 + s_2^2)(\tilde{s}_1^2 - \tilde{s}_2^2) + \frac{1}{2}(s_1^2 - s_2^2)(\tilde{s}_1^2 - \tilde{s}_2^2)$, where $\{\tilde{s}_1, \tilde{s}_2\} = \mathcal{S}(X)$.*

*Proof.* First note that since $X$ is assumed to be rank 2, $x \neq \alpha y$ and so the expression in the objective is exactly the numerator of the optimal value of the minimization problem over $W$ as stated in Lemma 6 (the denominator is constant within the scope of this lemma). Now, using the well-known formulas for the sum and product of squared singular values, we can easily rephrase our constraint in the following way:

$$a^T a + b^T b = s_1^2 + s_2^2 \quad \wedge$$
$$a^T a b^T b - (a^T b)^2 = s_1^2 s_2^2.$$

First, noticing that both the constraints as well as the objective only depend on $a^T a, b^T b, a^T b$, we can reduce the problem to one, where we treat these quantities as free variables that we are optimizing for. The only thing we need to remember will be implicit constraints that $a^T a, b^T b \geq 0$ and $(a^T b)^2 \leq a^T a b^T b$, which is necessary from Cauchy-Schwarz inequality. We will solve this optimization problem using, again, KKT conditions. This time, we will use a stronger rule for deciding on the necessity of KKT conditions – the linear independence constraint qualification (LICQ) rule. According to this rule, any optimal solution has to fulfill the KKT conditions if the gradients of the equality constraints (in this case only two) are linearly independent. In our case, if we order the variables as $(a^T a, b^T b, a^T b)$ and writing the gradients as: $(1, 1, 0)$ and $(b^T b, a^T a, -2a^T b)$ we see that they can only be dependent if $a^T b = 0$ and $a^T a = b^T b$. This is only a feasible solution if $s_1 = s_2$. We will return to this case later, showing that it achieves the same objective value as the one coming from KKT conditions.

Now, let us write down the Lagrange function to this problem:

$$\mathcal{L}(a^T a, b^T b, a^T b, u, v) = a^T a y^T y - 2a^T b x^T y + b^T b x^T x + u(a^T a + b^T b - s_1^2 - s_2^2)$$
$$+ v(a^T a b^T b - (a^T b)^2 - s_1^2 s_2^2).$$

Computing the partial derivatives we get:

$$\frac{\partial \mathcal{L}}{\partial a^T a} = y^T y + u + v b^T b \overset{!}{=} 0,$$
$$\frac{\partial \mathcal{L}}{\partial b^T b} = x^T x + u + v a^T a \overset{!}{=} 0,$$
$$\frac{\partial \mathcal{L}}{\partial a^T b} = -2x^T y - 2v a^T b \overset{!}{=} 0.$$

First consider what would happen if $v = 0$. In this case, $x^T y = 0$ from the third equation and $x^T x = y^T y$ from the first two. Then, however, the objective function is constant on the feasible set and equals $1/2(s_1^2 + s_2^2)(\tilde{s}_1^2 + \tilde{s}_2^2)$, which agrees with the statement of the Lemma.

Next, assume $v \neq 0$. Then we can express all the variables as follows:

$$a^T a = \frac{-x^T x - u}{v},$$
$$b^T b = \frac{-y^T y - u}{v},$$
$$a^T b = -\frac{x^T y}{v}.$$

Plugging this back into the constraints, we can solve for $v, u$. We get:

$$v = -\frac{x^T x + y^T y + 2u}{s_1^2 + s_2^2}.$$

Plugging this into the second constraint together with the expressions for $a^T a, b^T b$ and then simplifying we get:

$$\frac{(y^T y + u)(x^T x + u)(s_1^2 + s_2^2)^2}{(x^T x + y^T y + 2u)^2} - \frac{(x^T y)^2 (s_1^2 + s_2^2)^2}{(x^T x + y^T y + 2u)^2} = s_1^2 s_2^2$$

$$y^T y x^T x - (x^T y)^2 + (y^T y + x^T x)u + u^2 = \left(\frac{s_1 s_2}{s_1^2 + s_2^2}\right)^2 (x^T x + y^T y)^2$$

$$+ \left(\frac{2 s_1 s_2}{s_1^2 + s_2^2}\right)^2 (x^T x + y^T y)u + \left(\frac{2 s_1 s_2}{s_1^2 + s_2^2}\right)^2 u^2$$

$$\frac{(s_1^2 - s_2^2)^2}{(s_1^2 + s_2^2)^2} u^2 + \frac{(s_1^2 - s_2^2)^2}{(s_1^2 + s_2^2)^2}(x^T x + y^T y)u + x^T x y^T y - (x^T y)^2 - \left(\frac{s_1 s_2}{s_1^2 + s_2^2}\right)^2 (x^T x + y^T y)^2 = 0$$

$$u^2 + (x^T x + y^T y)u + \frac{(s_1^2 + s_2^2)^2}{(s_1^2 - s_2^2)^2}(x^T x y^T y - (x^T y)^2) - \left(\frac{s_1 s_2}{s_1^2 - s_2^2}\right)^2 (x^T x + y^T y)^2 = 0. \quad (13)$$

Recall that the case $s_1 = s_2$ is treated separately, so we do not divide by zero. Now we can rewrite this expression w.r.t. $\{\tilde{s}_1, \tilde{s}_2\}$ :

$$u^2 + (\tilde{s}_1^2 + \tilde{s}_2^2)u + \frac{(s_1^2 + s_2^2)^2}{(s_1^2 - s_2^2)^2}\tilde{s}_1^2 \tilde{s}_2^2 - \frac{s_1^2 s_2^2}{(s_1^2 - s_2^2)^2}(\tilde{s}_1^2 + \tilde{s}_2^2)^2 = 0$$

We can now simply solve for this to obtain the roots:

$$u_\pm = \frac{-(\tilde{s}_1^2 + \tilde{s}_2^2) \pm \sqrt{(\tilde{s}_1^2 + \tilde{s}_2^2)^2 + \frac{4 s_1^2 s_2^2}{(s_1^2 - s_2^2)^2}(\tilde{s}_1^2 + \tilde{s}_2^2)^2 - 4\frac{(s_1^2 + s_2^2)^2}{(s_1^2 - s_2^2)^2}\tilde{s}_1^2 \tilde{s}_2^2}}{2},$$

which after some simplifications yields:

$$u_\pm = \frac{-(\tilde{s}_1^2 + \tilde{s}_2^2) \pm \frac{s_1^2 + s_2^2}{s_1^2 - s_2^2}(\tilde{s}_1^2 - \tilde{s}_2^2)}{2}.$$

The formula for $v$ can now be obtained:

$$v_\pm = \mp\frac{\tilde{s}_1^2 - \tilde{s}_2^2}{s_1^2 - s_2^2}$$

Having this, we can now compute the objective value using these quantities. We write it down for $u_+, v_+$ :

$$b^T b x^T x + a^T a y^T y - 2 a^T b x^T y = \frac{-(x^T x + y^T y)u - 2(x^T x y^T y - (x^T y)^2)}{v}$$

$$= \frac{-(\tilde{s}_1^2 + \tilde{s}_2^2)u - 2\tilde{s}_1^2 \tilde{s}_2^2}{v} = \frac{(\tilde{s}_1^2 + \tilde{s}_2^2)^2 - \frac{s_1^2 + s_2^2}{s_1^2 - s_2^2}(\tilde{s}_1^2 - \tilde{s}_2^2)(\tilde{s}_1^2 + \tilde{s}_2^2) - 4\tilde{s}_1^2 \tilde{s}_2^2}{2v}$$

$$= \frac{-\frac{(\tilde{s}_1^2 + \tilde{s}_2^2)^2}{\tilde{s}_1^2 - \tilde{s}_2^2}(s_1^2 - s_2^2) + (s_1^2 + s_2^2)(\tilde{s}_1^2 + \tilde{s}_2^2) + 4\tilde{s}_1^2 \tilde{s}_2^2 \frac{s_1^2 - s_2^2}{\tilde{s}_1^2 - \tilde{s}_2^2}}{2} = \frac{(s_1^2 + s_2^2)(\tilde{s}_1^2 + \tilde{s}_2^2) - (s_1^2 - s_2^2)(\tilde{s}_1^2 - \tilde{s}_2^2)}{2},$$

which is the expression we wanted to prove. Had we done the same computation, but for $u_-, v_-$, we would get

$$\frac{(s_1^2 + s_2^2)(\tilde{s}_1^2 + \tilde{s}_2^2) + (s_1^2 - s_2^2)(\tilde{s}_1^2 - \tilde{s}_2^2)}{2},$$

which is at least as big as the previous expression, so we can drop this one. However, we still need to check, whether this solution of the KKT system is indeed feasible, because we imposed implicit constraints into the problem. Thus, we need to check: $a^T a, b^T b \geq 0$ and $a^T a b^T b \geq (a^T b)^2$. The first two are analogous. We have: $a^T a = -\frac{x^T x + u}{v}$, but since the optimal $v_+$ is strictly negative, we can only analyze the sign of $2 x^T x + 2u$ which is:

$$2 x^T x + 2u = x^T x - y^T y + \frac{s_1^2 + s_2^2}{s_1^2 - s_2^2}(\tilde{s}_1^2 - \tilde{s}_2^2) \geq \tilde{s}_1^2 - \tilde{s}_2^2 - |x^T x - y^T y| \overset{?}{\geq} 0.$$

However, using the same formulas as the constraints in this optimization problem on $\tilde{s}_1, \tilde{s}_2$ instead, one can, using very simple computations, check that this indeed holds. For the next condition, consider that $a^T a b^T b - (a^T b)^2$ must have the same sign as $(x^T x + u)(y^T y + u) - (x^T y)^2 = x^T x y^T y - (x^T y)^2 + u^2 + u(x^T x + y^T y)$. This resembles the quadratic function in (13). $u_+$ is a root of that quadratic function. Both this new quadratic function, as well as the one in (13) have the same coefficients in front of $u^2$ and $u$. Therefore, to test for the sign of the expression above, it only suffices to check, whether this alternative quadratic function has at least as big absolute constant:

$$x^T x y^T y - (x^T y)^2 - \frac{(s_1^2 + s_2^2)^2}{(s_1^2 - s_2^2)^2}(x^T x y^T y - (x^T y)^2) + \left(\frac{s_1 s_2}{s_1^2 - s_2^2}\right)^2 (x^T x + y^T y)^2 \geq 0 \iff$$

$$-\frac{s_1^2 s_2^2}{(s_1^2 - s_2^2)^2} 4\tilde{s}_1^2 \tilde{s}_2^2 + \frac{s_1^2 s_2^2}{(s_1^2 - s_2^2)^2}(\tilde{s}_1^2 + \tilde{s}_2^2)^2 \geq 0 \iff \frac{s_1^2 s_2^2}{(s_1^2 - s_2^2)^2}(\tilde{s}_1^2 - \tilde{s}_2^2)^2 \geq 0.$$

The last inequality trivially holds. Therefore, the solution of KKT conditions is feasible.

Now, let us consider the special case when $a^T b = 0$, $a^T a = b^T b$ and $s_1 = s_2$ for which the LICQ criterion is not satisfied. A very simple computation then yields that any feasible solution achieves the same objective value as the one above. This means that if the problem does have a global solution, it must be this one, otherwise the optimal solution would neither satisfy a KKT condition, nor be in the region where LICQ does not hold. Therefore, we must, finally, reason that this is indeed a globally optimal solution, which is not guaranteed by KKT conditions themselves. For this, it is only necessary to note that the feasible set is obviously compact. Since the objective value is continuous, we can use the well-known fact that such an optimization problem indeed has a global solution. $\quad\square$

Now we are ready to finish the proof of Lemma 5. Let us first still assume $\tilde{s}_1, \tilde{s}_2 > 0$. Then, combining Lemma 6 with Lemma 7 we get:

$$\min_{W} \quad \|W\|_F^2$$
$$\text{s.t.} \quad \mathcal{S}(\sigma(WX)) = \{s_1, s_2\}$$

equals

$$\frac{(s_1^2 + s_2^2)(\tilde{s}_1^2 + \tilde{s}_2^2) - (s_1^2 - s_2^2)(\tilde{s}_1^2 - \tilde{s}_2^2)}{2\tilde{s}_1^2 \tilde{s}_2^2} = \frac{s_1^2}{\tilde{s}_1^2} + \frac{s_2^2}{\tilde{s}_2^2}.$$

Now consider the case $\tilde{s}_2 = 0$. According to Lemma 6 this implies $s_2 = 0$. When optimizing over the matrix $A$ we are facing the following optimization problem:

$$\min_{A} \quad \frac{b^T b}{y^T y}$$
$$\text{s.t.} \quad \mathcal{S}(A) = \{s_1, 0\},$$
$$\frac{a^T a}{b^T b} = \frac{x^T x}{y^T y},$$

where the second constraint is clear from the considerations done in Lemma 6. Combining these two constraints, one simply gets

$$b^T b = \frac{y^T y}{x^T x + y^T y} s_1^2$$

and plugging this in the objective value we have $\frac{s_1^2}{\tilde{s}_1^2}$, which is consistent with the statement of Lemma 7. The case when $X$ is 0 is trivial. This proves Lemma 5. $\quad\square$

Notice that, for all indices but $l = 1$, we managed to reduce the optimization over full matrices $W_l$ to the optimization over only the singular values of $\sigma(H_k)$ matrices. We will do the same with the first layer's matrices $W_1, H_1$. For this, we again split the optimization of $\frac{\lambda_{W_1}}{2}\|W_1\|_F^2 + \frac{\lambda_{H_1}}{2}\|H_1\|_F^2$ given $\{s_1, s_2\} = \mathcal{S}(\sigma(W_1 H_1))$ into two nested sub-problems, the inner one optimizing the quantity only for a fixed output $H_2 := W_1 H_1$ and the outer one optimizing over the output $H_2$ given the optimal value of the inner problem. We can solve the inner subproblem by a direct use of the following lemma, which describes a variational form of a nuclear norm.

**Lemma 8.** *The optimization problem*

$$\min_{A,B;C=AB} \quad \frac{\lambda_A}{2} \|A\|_F^2 + \frac{\lambda_B}{2} \|B\|_F^2 \tag{6}$$

*is minimized at value $\sqrt{\lambda_A \lambda_B} \|C\|_*$, and the minimizers are of the form $A^* = \gamma_A U \Sigma^{1/2} R^T$, $B^* = \gamma_B R \Sigma^{1/2} V^T$. Here, the constants $\gamma_A, \gamma_B$ only depend on $\lambda_A, \lambda_B$; $U\Sigma V^T$ is the SVD of $C$; and $R$ is an orthogonal matrix.*

*Proof.* See Lemma C.1 of [28]. □

Using this Lemma 8, we know that the optimal value of the inner optimization problem is $\sqrt{\lambda_{W_1} \lambda_{H_1}} \|H_2\|_*$. Therefore we only need to solve the outer problem:

$$\min_{H_2} \|H_2\|_*$$
$$\text{s.t.} \quad \mathcal{S}(\sigma(H_2)) = \{s_1, s_2\}.$$

To solve this problem, we will state a much more general statement, which can be of independent interest, in the following lemma.

**Lemma 9.** *Let $L \geq 2$ be a positive integer. Then, the optimal value of the optimization problem*

$$\min_H \|H\|_{S_{\frac{2}{L}}}^{\frac{2}{L}}, \qquad\qquad \text{s.t.} \quad \mathcal{S}(\sigma(H)) = \{s_1, s_2\} \tag{7}$$

*equals $(s_1 + s_2)^{\frac{2}{L}}$. Here $\|\cdot\|_{S_p}$ denotes the $p$-Schatten pseudo-norm.*

*Proof.* First note that this optimization problem has two free components: *(i)* $\sigma(H)$ so that it satisfies the constraint, and *(ii)* $H$ with $\sigma(H)$ being fixed. This enables us to split the problem into inner and outer minimization. However, the complexity of the problem will prevent us from doing this conditioning in a straightforward way. Further, let us also implicitly note that within the proof we implicitly assume the matrix in question has at least three rows. The case with two rows is very simple and follows the proof with more rows.

We start by explicitly computing the objective value as a function of important statistics of $H$. Let $\sigma(H)$ have columns $r, s$. Then,

$$\sigma(H)^T \sigma(H) = \begin{pmatrix} r^T r & r^T s \\ r^T s & s^T s \end{pmatrix}.$$

Using the well-known formula on the eigenvalues of such matrix, we know $\lambda_1 + \lambda_2 = r^T r + s^T s$ and $\lambda_1 \lambda_2 = r^T r s^T s - (r^T s)^2$. Solving this system we get:

$$s_{1,2}^2 = \lambda_{1,2} = \frac{r^T r + s^T s \pm \sqrt{(r^T r + s^T s)^2 - 4(r^T r s^T s - (r^T s)^2)}}{2}. \tag{14}$$

Let us, for now, fix this $\sigma(H)$. For a while we will only optimize over $H$ s.t. $\sigma(H)$ is fixed. We will think about the optimization problem in the following way: consider any $\delta \geq 0$. Assume we are allowed to "inject" negative values in the place of zero entries of a fixed $\sigma(H)$ so that the sum of squares of those negative entries is exactly $\delta$. What is the minimal value of the objective value given this $\delta$ and what is the optimal injection? Let us denote any quantity which was subject to such injection (or perturbation) with an extra lower-index $P$.

To answer this question, first have a closer look at (14). It is clear that after injecting the negative noise of total squared size $\delta$, the $r_P^T r_P + s_P^T s_P$ in a perturbed matrix will increase by exactly $\delta$ compared to $r^T r + s^T s$. Therefore, if we condition on $\delta$, the only degree of freedom lies in the $-4(r_P^T r_P s_P^T s_P - (r_P^T s_P)^2)$ in the perturbed matrix. Crucially, since the objective value is $\lambda_{1,P}^{1/L} + \lambda_{2,P}^{1/L}$, by concavity of $1/L$-th power, for any fixed sum of $\lambda_{1,P} + \lambda_{2,P} = r^T r + s^T s + \delta$ we want the $\lambda_{i,P}$ to be as far apart as possible. This is obtainable by maximizing the quantity $-4(r_P^T r_P s_P^T s_P - (r_P^T s_P)^2)$ which is equivalent to minimizing $r_P^T r_P s_P^T s_P - (r_P^T s_P)^2$. Importantly, as it will be clear from the forthcoming reasoning, the minimal value of $\lambda_{2,P}$ is decreasing with

$\delta$. Therefore, it only makes sense to consider all the $\delta$ values for which the minimal $\lambda_{2,P}$ is still bigger than 0 up to the $\delta$ value for which the minimal $\lambda_{2,P}$ hits 0 for the first time. From that $\delta$ on, every optimal $\lambda_{2,P}$ is 0, but the $\lambda_{1,P}$ increases and thus becomes sub-optimal. Now let us split the $\delta$ into $\delta_1$ and $\delta_2$, denoting the total squared size of perturbations per column. If we also fix $\delta_1$ and $\delta_2$ and condition on them, even the quantity $r_P^T r_P s_P^T s_P$ is fixed and we are left with maximizing $(r_P^T s_P)^2$. Denote $p = r^T s$. To make the next argument clear, let us split the matrix $\sigma(H)$ into four blocks of rows, where in each block there is one of the four following types of rows: $(+,+), (+,0), (0,+), (0,0)$. We will now define variables $a, b, c, d$ as sum of squared entries corresponding to one column of one block. Specifically:

$$a = \sum_{i;\sigma(H)_{i1}>0,\sigma(H)_{i2}>0} \sigma(H)_{i1}^2; \quad b = \sum_{i;\sigma(H)_{i1}>0,\sigma(H)_{i2}>0} \sigma(H)_{i2}^2$$

$$c = \sum_{i;\sigma(H)_{i1}>0,\sigma(H)_{i2}=0} \sigma(H)_{i1}^2; \quad d = \sum_{i;\sigma(H)_{i1}=0,\sigma(H)_{i2}>0} \sigma(H)_{i2}^2.$$

The value we are trying to minimize equals, for an unperturbed $\sigma(H)$ written in this way, $(a + c)(b + d) - p^2$. Of course, $\sigma(H)$ might not have all the types of blocks, in which case the empty sum is just defined as 0. First, we argue that whatever is the optimal assignment of the negative values given $\delta_1, \delta_2$, they do not appear in both the $(0,0)$ block and $(+,0), (0,+)$ blocks. This is because the only variable we are optimizing for in this conditioning is $(r_P^T s_P)^2$. Assume that, given the perturbation, the quantity $r_P^T s_P$ is negative. Then, any negative values present in the $(0,0)$ block can be moved to $(+,0), (0,+)$ blocks strictly decreasing $r_P^T s_P$ and thus increasing $(r_P^T s_P)^2$. On the other hand, if $r_P^T s_P$ happens to be positive, then any negative values present in the $(+,0), (0,+)$ blocks can be moved to the $(0,0)$ block, strictly increasing the $r_P^T s_P$ value and thus $(r_P^T s_P)^2$ too. Therefore, we only need to consider cases when we inject the negative values to either the $(0,0)$ block or to $(+,0), (0,+)$ blocks, but not both at once. Let us start with the second case and analyze it. Clearly, if $\sigma(H)$ did not have either the $(0,0)$ block or $(+,0), (0,+)$ blocks, then we do not even need this reasoning anymore. Also it is clear, as we will see later, that a matrix $\sigma(H)$ which would completely lack 0 entries is suboptimal and thus there is always space to inject negative entries.

**Case 1: Injecting negative values to $(+,0), (0,+)$ blocks.** For the same definition of $a, b, c, d$ as above, we further define the squared sizes of total injected negative values in the perturbed $H_P$ per block as $x, y$, i.e.,

$$x = \sum_{i;\sigma(H)_{i1}>0,\sigma(H)_{i2}=0} H_{P,i2}^2; \quad y = \sum_{i;\sigma(H)_{i1}=0,\sigma(H)_{i2}>0} H_{P,i1}^2,$$

and thus $x = \delta_2, y = \delta_1, x + y = \delta$. Trivially, such a construction only makes sense if it is possible to make $r_P^T s_P$ negative, otherwise it is strictly better to construct an alternative matrix $\overline{\sigma(H)}$ with $a + c = \overline{a} + \overline{c}; b + d = \overline{b} + \overline{d}; p = \overline{p}$, but which includes the $(0,0)$ block and put all the negative value there (this is always possible for matrices with at least 3 rows, since we only need two rows to construct $\overline{\sigma(H)}$ which satisfies the outlined constraints). This alternative matrix together with negative values in the $(0,0)$ block would be better, because obviously $(\overline{r_P}^T \overline{s_P})^2 > (r_P^T s_P)^2$.

Once $r_P^T s_P$ is negative, though, it is clear that it is optimal, given the resources $x, y$, to produce as negative value as possible so as to maximize $(r_P^T s_P)^2$. By the use of Cauchy-Schwartz inequality, this is possible exactly when the injected negative vectors in the blocks are aligned with their positive counterparts of that block. Then, the inner product between the negative and positive part of a block will equal exactly $\sqrt{cx}$ or $\sqrt{yd}$. Furthermore, if it is possible to construct $\overline{\sigma(H)}$ for which $\overline{a}+\overline{b} < a+b; \overline{a}+\overline{c} = a+c; \overline{b}+\overline{d} = b+d; \overline{p} = p$ (which is only possible if there are at least two rows in the $(+,+)$ block and if $p^2 < ab$ – thanks to the fact that then the angle between columns in the $(+,+)$ block is a free variable and we can decrease it while decreasing $a, b$ to keep $p$ unchanged using the law of cosine), then our objective value strictly decreases, since $\sqrt{\overline{c}x} + \sqrt{\overline{y}\overline{d}} > \sqrt{cx} + \sqrt{dy}$. Therefore, it is always optimal for the first and second column of the $(+,+)$ block to be aligned, which also yields $ab = p^2$. This value of $p$ is necessary also if the $(+,+)$ block has only one row. Thus, we have that the original value of $r^T r s^T s - (r^T s)^2$ is $(a + c)(b + d) - ab$. The value after perturbation, using all the knowledge we acquired so far is $(a+c+y)(b+d+x) - (\sqrt{ab} - \sqrt{cx} - \sqrt{dy})^2$. Subtracting, we get a difference: $by + ax + xy + 2\sqrt{abc}\sqrt{x} + 2\sqrt{abd}\sqrt{y} - 2\sqrt{cd}\sqrt{xy}$. Consider now an alternative

matrix $\overline{H_P}$ such that $x, y$ remain unchanged and moreover $\overline{a} = \overline{b} = 0$, $\overline{c} + \overline{d} = a + b + c + d$ and $\overline{cd} = (a+c)(b+d) - ab$, meaning that the $\sigma(\overline{H_P})$ has the same singular values as $\sigma(H_P)$. However, computing the same difference of the same quantities one gets: $xy - 2\sqrt{\overline{cd}}\sqrt{xy}$. This value is strictly smaller than the previous one for any fixed pair of $x, y$. Therefore, a matrix $\sigma(H)$ which *does not* contain the $(+, +)$ block is the only possible optimal solution for the injection which goes into $(+, 0), (0, +)$ blocks. Thus, we can reduce the previous problem into considering a matrix $H_P$ with $a = b = 0$ and minimizing the difference in objective values $xy - 2\sqrt{cd}\sqrt{xy}$, given $x + y = \delta$. After plugging in $y = \delta - x$, differentiating for $x$ and putting equal to 0, we get $\delta - 2x - \frac{\sqrt{cd}}{\sqrt{x(\delta-x)}}(\delta - 2x) = 0$, which is equivalent to $(\delta - 2x)(1 - \frac{\sqrt{cd}}{\sqrt{x(\delta-x)}}) = 0$. From here, we see that either $x = \delta/2$ or $cd = x(\delta - x)$, which after solving yields $x_{1,2} = \frac{\delta \pm \sqrt{\delta^2 - 4cd}}{2}$. If $\delta^2 \le 4cd$ then the only stationary point is $\delta/2$ and this is clearly an optimal solution. For $\delta^2 = 4cd$, the optimal solution yields a rank 1 matrix so that $\lambda_{2,P} = 0$. This means we do not need to care about $\delta^2 > 4cd$, as this would not achieve smaller value of $\lambda_{2,P}$, but would increase the value of $\lambda_{1,P}$. So, the solution where $x = y = \delta/2$ is optimal and the objective value is $\delta^2/4 - s_1 s_2 \delta$. This is clearly smaller than 0 so we actually achieved a decrease in objective value.

**Case 2: Injecting negative values to $(0,0)$ block.** For this, it is not necessary to distinguish $(+,+),(+,0),(0,+)$ blocks, so we treat them as one. We can define:

$$e = \sum_{i;\sigma(H)_{i1}+\sigma(H)_{i2}>0} \sigma(H)_{i1}^2 +; \qquad f = \sum_{i;\sigma(H)_{i1}+\sigma(H)_{i2}>0} \sigma(H)_{i2}^2$$

$$x = \sum_{i;\sigma(H)_{i1}=0,\sigma(H)_{i2}=0} \sigma(H)_{i1}^2; \qquad y = \sum_{i;\sigma(H)_{i1}=0,\sigma(H)_{i2}=0} \sigma(H)_{i2}^2.$$

The original objective value is $ef - p^2$, while the perturbed one is $(e+x)(f+y) - (p + \sqrt{xy})^2$ and so the difference is $xf + ye - 2p\sqrt{xy}$. However, using Cauchy-Schwartz inequality, this is bigger or equal to $xf + ye - 2\sqrt{fexy} \ge 0$. Thus, it is not possible to improve the objective value in this case, unlike the previous case where it was. Therefore, this case is clearly sub-optimal and we do not need to deal with it.

The only thing that could potentially go wrong with this proof is that there is not enough rows to build our necessary blocks. This can only happen if $\sigma(H)$ is a $2 \times 2$ matrix. This case is, however, very simple and we will not explicitly treat it, but the statement and the proof path are the same for it.

We solved our optimization problem for every fixed allowed perturbation size $\delta$. Now we just need to optimize over $0 \le \delta \le 2s_1 s_2$. So we face the following optimization problem:

$$\min_{0 \le \delta \le 2s_1 s_2} \left( \frac{s_1^2 + s_2^2 + \delta}{2} + \frac{1}{2}\sqrt{(s_1^2 + s_2^2 + \delta)^2 - 4(s_1^2 s_2^2 + \delta^2/4 - s_1 s_2 \delta)} \right)^{\frac{1}{L}}$$

$$+ \left( \frac{s_1^2 + s_2^2 + \delta}{2} - \frac{1}{2}\sqrt{(s_1^2 + s_2^2 + \delta)^2 - 4(s_1^2 s_2^2 + \delta^2/4 - s_1 s_2 \delta)} \right)^{\frac{1}{L}}.$$

Denote $\lambda_{\delta,1}$ the value of the first summand of the above objective value (without $1/L$-th power) and similarly $\lambda_{\delta,2}$ the value of the second summand (so those are the eigenvalues of optimal $H_P^T H_P$ given exactly $\delta$ allowed negative injection). The derivative of the objective w.r.t. $\delta$ equals:

$$\frac{1}{n\lambda_{\delta,1}^{\frac{L-1}{L}}} \left( \frac{1}{2} + \frac{(s_1 + s_2)^2}{2D_\delta} \right) + \frac{1}{n\lambda_{\delta,2}^{\frac{L-1}{L}}} \left( \frac{1}{2} - \frac{(s_1 + s_2)^2}{2D_\delta} \right),$$

where $D_\delta := \sqrt{(s_1^2 + s_2^2 + \delta)^2 - 4(s_1^2 s_2^2 + \delta^2/4 - s_1 s_2 \delta)}$ denotes the square root part of the $\lambda_{\delta;1,2}$ expression. We will prove that this derivative is either 0 for $L = 2$ or strictly negative on $[0, 2s_1 s_2]$ everywhere except a set of measure 0 for $L \ge 3$. This guarantees for $L = 2$ the optimal value being equal to the unperturbed matrix while also defines the set of optimal solutions. For $L \ge 3$ the optimal value is attained at $\delta = 2s_1 s_2$, the objective value is easily expressible and the optimal solution is characterized by a single perturbation size. To test whether this expression is smaller than 0, nothing happens if we multiply it with $2L\lambda_{\delta,1}^{\frac{L-1}{L}} \lambda_{\delta,2}^{\frac{2L-2}{L}}$, so we get:

$$\lambda_{\delta,1}^{\frac{2L-2}{L}} \left( 1 + \frac{(s_1 + s_2)^2}{D_\delta} \right) + \lambda_{\delta,1}^{\frac{L-1}{L}} \lambda_{\delta,2}^{\frac{L-1}{L}} \left( 1 - \frac{(s_1 + s_2)^2}{D_\delta} \right).$$

We can now compute

$$\lambda_{\delta,1}\lambda_{\delta,2} = \frac{(s_1^2 + s_2^2 + \delta)^2 - (s_1^2 - s_2^2)^2 - 2(s_1 + s_2)\delta}{4} = \left(\frac{2s_1 s_2 - \delta}{2}\right)^2$$

to get

$$\lambda_{\delta,2}^{\frac{2L-2}{L}}\left(1 + \frac{(s_1 + s_2)^2}{D_\delta}\right) + \left(\frac{2s_1 s_2 - \delta}{2}\right)^{\frac{2L-2}{L}}\left(1 - \frac{(s_1 + s_2)^2}{D_\delta}\right).$$

This expression resembles a form for which the use of Jensen inequality would be convenient. However, there are two issues. First, the coefficients, though summing up to 1 are not a convex combination since $\frac{(s_1+s_2)^2}{D_\delta} \geq 1$. So this is a combination outside of the convex hull of the inputs. However, Jensen inequality outside the convex hull is applicable with opposite sign. The bigger issue is that the function to use the Jensen inequality on, $f(x) = x^{\frac{2L-2}{L}}$ is only easily defined on $\mathbb{R}_+$. To make sure that we can safely use Jensen inequality, we first need to check that the linear combination, though outside of the convex hull of the inputs, still lies in $\mathbb{R}_+$. We are therefore asking whether

$$\lambda_{\delta,2}\left(1 + \frac{(s_1 + s_2)^2}{D_\delta}\right) + \frac{2s_1 s_2 - \delta}{2}\left(1 - \frac{(s_1 + s_2)^2}{D_\delta}\right) \geq 0.$$

Now, we can multiply out and simplify this expression (after multiplying by 2):

$$s_1^2 + s_2^2 + \delta - D_\delta + (s_1^2 + s_2^2)\frac{(s_1 + s_2)^2}{D_\delta} + \delta\frac{(s_1 + s_2)^2}{D_\delta} - (s_1 + s_2)^2 + 2s_1 s_2$$

$$- \delta - \frac{2s_1 s_2(s_1 + s_2)^2}{D_\delta} + \frac{\delta(s_1 + s_2)^2}{D_\delta} \geq 0 \iff$$

$$((s_1 - s_2)^2 + 2\delta)(s_1 + s_2)^2 \geq (s_1^2 - s_2^2)^2 + 2(s_1 + s_2)^2\delta,$$

but this holds even with equality. So we proved that the combination of inputs we are considering here is exactly 0 for which $x^{\frac{2L-2}{L}}$ is well-defined. So we can freely use Jensen inequality:

$$\lambda_{\delta,2}^{\frac{2L-2}{L}}\left(1 + \frac{(s_1 + s_2)^2}{D_\delta}\right) + (2s_1 s_2 - \delta)^{\frac{2L-2}{L}}\left(1 - \frac{(s_1 + s_2)^2}{D_\delta}\right) \leq$$

$$\left(\lambda_{\delta,2}\left(1 + \frac{(s_1 + s_2)^2}{D_\delta}\right) + \frac{2s_1 s_2 - \delta}{2}\left(1 - \frac{(s_1 + s_2)^2}{D_\delta}\right)\right)^{\frac{2L-1}{L}} = 0.$$

Note that if $L = 2$, the function we applied Jensen inequality on was, in fact, linear and thus we retained equality. If $L \geq 3$, then since the function in question is strictly convex, the equality can only hold if $\lambda_{\delta,2} = 2s_1 s_2 - \delta$. After writing down the expression on $\lambda_{\delta,2}$ it becomes clear that this is equivalent to a quadratic equation on $\delta$, which can have at most two solutions. So we proved that the derivative is smaller than 0 on the whole interval of interest except a set of measure at most 0, where it is 0. Therefore, the only optimum is at $\delta = 2s_1 s_2$. Plugging this in the objective value one gets $(s_1 + s_2)^{\frac{2}{L}}$ and from the proof we also get the precise description of the optimizers for both $L = 2$ and $L > 2$ cases. $\qquad\square$

Now we are ready to formally make the reduction of the optimization problem in (10) into the optimization over only singular values of the matrices $\sigma(H_l)$ for $2 \leq l \leq L$. We formulate this reduction in a separate lemma for clarity.

**Lemma 12.** *The optimal value of the following optimization problem:*

$$\min_{W_L,\ldots,W_1,H_1} \frac{1}{4}\|W_L\sigma(W_{L-1}\ldots W_2\sigma(W_1 H_1)) - I_2\|_F^2 + \sum_{l=1}^{L}\frac{\lambda_{W_l}}{2}\|W_l\|_F^2 + \frac{\lambda_{H_1}}{2}\|H_1\|_F^2$$

$$s.t. \quad \{s_{l,1}; s_{l,2}\} = \mathcal{S}(\sigma(H_l)) \ \forall\, 2 \leq l \leq L$$

*with $H_1$ having two columns and all the matrices having at least two rows is*

$$\sum_{i=1}^{2}\left(\frac{\lambda_{W_L}}{2(s_{L,i}^2 + 2\lambda_{W_L})} + \sum_{l=2}^{L-1}\left(\frac{\lambda_{W_l}}{2}\frac{s_{l+1,i}^2}{s_{l,i}^2}\right) + \sqrt{\lambda_{W_1}\lambda_{H_1}}s_{2,i}\right),$$

*where the $\frac{s_{l+1,i}^2}{s_{l,i}^2}$ is defined as 0 if $s_{l+1,i}^2 = s_{l,i}^2 = 0$.*

*Proof.* The proof follows easily by applying the Lemmas 9, 5 and 4. □

Now we are only left with optimizing the objective over all the $s_{l,i}$. Crucially, the objective function is separable and symmetric in the $i$ index. Therefore we can optimize for each index separately. Importantly, if there is only a single solution to this problem, then the solution must be the same for both indices $i$, which will yield orthogonality of the $\sigma(H_l)$ matrices. The ultimate goal is to show that this is indeed the case modulo some special circumstances. The optimal value of this reduced optimization problem is presented in the following, final lemma with abstract notation of the free variables.

**Lemma 11.** *The optimization problem*

$$\min_{x_l; 2 \leq l \leq L} \frac{\lambda_{W_L}}{2(x_L + 2\lambda_{W_L})} + \sum_{l=2}^{L-1} \left( \frac{\lambda_{W_l}}{2} \frac{x_{l+1}}{x_l} \right) + \sqrt{\lambda_{W_1}\lambda_{H_1}}\sqrt{x_2} \tag{8}$$

*is optimized at an entry-wise positive, unique solution if*

$$\lambda_{H_1} \prod_{l=1}^{L} \lambda_{W_l} < \frac{(L-1)^{L-1}}{2^{L+1}L^{2L}}, \tag{9}$$

*and at $(x_2, \ldots, x_L) = (0, \ldots, 0)$ otherwise. When (9) holds with equality, both solutions are optimal. Here, $0/0$ is defined to be $0$.*

*Proof.* Here we will need to distinguish the cases $L = 2$ and $L > 2$. We start with the case $L = 2$. Let us reparametrize the problem so that we will work with $y = \sqrt{x_2}$. We get the following expression to optimize the $y$ over:

$$\frac{\lambda_{W_2}}{2(y^2 + 2\lambda_{W_2})} + \sqrt{\lambda_{W_1}\lambda_{H_1}}\, y.$$

This function is simple enough so that we understand its shape. It has precisely one inflection point, below which it is concave, later it is convex. The derivative at $0$ is strictly positive. Judging from this, the function either has $0$ as a single global optimum, or it has two global optima, one at $0$ and one at a strictly positive local minimum of the strictly convex part of the function, or the global minimum is achieved uniquely at the strictly convex part of the function. To find out, we only need to solve $f(y) = f(0)$ for $y$, where $f$ stands for the optimized function and determine the number of solutions. The quadratic equation this is equivalent to after excluding the point $y = 0$ as a solution is $4y^2\sqrt{\lambda_{W_1}\lambda_{H_1}} - y + 8\lambda_{W_2}\sqrt{\lambda_{W_1}\lambda_{H_1}} = 0$. This has a unique solution if and only if $\lambda_{W_2}\lambda_{W_1}\lambda_{H_1} = 1/256$, which is precisely the threshold from the statement of the lemma. The cases outside of this threshold are easily mapped to the system having the global solution $0$ or some positive point.

Let us now consider the case $L > 2$. First note that, once for any $l_0$ the optimal $x_{l_0}^* = 0$, then at this particular solution all the $x$'s must be $0$. If $l_0 = L$, then this claim is apparent backward-inductively optimizing always over $x$ with one index lower. If $2 \leq l_0 < L$, then the claim is trivial backward-inductively for all $l \leq l_0$, while for all $l > l_0$ this solution is forced by feasibility issues. From now on we will thus work with an implicit strict positiveness assumption. Taking derivatives of this objective w.r.t. $x_l$ for different $l$ and putting them equal $0$ we arrive at three types of equations depending on $l$:

$$\sqrt{\frac{\lambda_{W_L}}{\lambda_{W_{L-1}}}}\sqrt{x_{L-1}} = x_L + 2\lambda_{W_L},$$

$$x_l^2 = \frac{\lambda_{W_l}}{\lambda_{W_{l-1}}} x_{l+1}x_{l-1}, \quad \text{for } 3 \leq l \leq L-1,$$

$$x_2^{\frac{3}{2}} = \frac{\lambda_{W_2}}{\sqrt{\lambda_{W_1}\lambda_{H_1}}} x_3. \tag{15}$$

Here we can notice two things. First, we have $L - 1$ equations for $L - 1$ variables, so we should expect a single solution in most of the cases. Second, the equations are rather simple and recursive.

Having either a value of $x_2$ or $x_L$, all other values are uniquely determined. Let us denote $q := \frac{x_L}{x_{L-1}}$. Based on this, we can arrive at the following formula:

$$x_{L-k} = \frac{\prod_{j=1}^{k-1} \lambda_{W_{L-j-1}}}{\lambda_{W_{L-1}}^{k-1}} q^{-k} x_L, \tag{16}$$

for all $1 \le k \le L - 2$. An empty product is defined as 1. To derive this, for $L = 3$ the statement is only about $x_2$ and is trivial. For $L > 3$, the $k = 1$ case is trivial, then one can proceed by induction using the recurrent formula $x_l^2 = \frac{\lambda_{W_l}}{\lambda_{W_{l-1}}} x_{l+1} x_{l-1}$. From formula (16), one can derive another useful equation:

$$\frac{x_{k+1}}{x_k} = \frac{\lambda_{W_{L-1}}}{\lambda_{W_k}} q, \qquad \text{for } 2 \le k \le L - 1.$$

Combining this with (15), we also get $x_2 = \frac{\lambda_{W_{L-1}}^2}{\lambda_{W_1} \lambda_{H_1}} q^2$. Combining now this equation with formula (16) for $x_2$ from before one gets:

$$\frac{\lambda_{W_{L-1}}^2}{\lambda_{W_1} \lambda_{H_1}} q^2 = \frac{\prod_{j=1}^{L-3} \lambda_{W_{L-j-1}}}{\lambda_{W_{L-1}}^{L-3}} q^{2-L} x_L \iff x_L = \frac{\lambda_{W_{L-1}}^{L-1}}{\lambda_{H_1} \prod_{j=1}^{L-2} \lambda_{W_j}} q^L.$$

Notice that we managed to express all the terms present in the main objective as simple functions of $q$ and regularization parameters. Plugging all of these expressions back in the objective, we get:

$$\frac{\lambda_{W_L}}{2 \left( \frac{\lambda_{W_{L-1}}^{L-1}}{\lambda_{H_1} \prod_{j=1}^{L-2} \lambda_{W_j}} q^L + 2\lambda_{W_L} \right)} + \frac{L \lambda_{W_{L-1}}}{2} q. \tag{17}$$

Since we only care about relative values of this expression, the optimization will be equivalent after multiplying it by $2/\lambda_{W_{L-1}}$ so we get:

$$\frac{1}{\frac{\lambda_{W_{L-1}}^L}{\lambda_{H_1} \lambda_{W_L} \prod_{j=1}^{L-2} \lambda_{W_j}} q^L + 2\lambda_{W_{L-1}}} + Lq. \tag{18}$$

Differentiating this function twice we can see that the second derivative is negative from 0 up to a single threshold at which it is 0, then it is positive. Therefore, based on the value of the first derivative at this threshold, and taking into account that the first derivative at 0 is strictly positive, we can easily see that the function has either a single global minimum at 0 or it has two global minima at 0 and at some positive $q$ or it has a single global optimum at some positive $q$. To find out which of the cases we are at, we can simply solve for

$$\frac{1}{\frac{\lambda_{W_{L-1}}^L}{\lambda_{H_1} \lambda_{W_L} \prod_{j=1}^{L-2} \lambda_{W_j}} q^L + 2\lambda_{W_{L-1}}} + Lq = \frac{1}{2\lambda_{W_{L-1}}}.$$

After excluding the trivial solution $q = 0$, this is equivalent to:

$$2\lambda_{W_{L-1}} L q^L - q^{L-1} + \frac{4L \lambda_{H_1} \lambda_{W_L} \prod_{j=1}^{L-2} \lambda_{W_j}}{\lambda_{W_{L-1}}^{L-2}} = 0.$$

The case at which we stand w.r.t. the global solutions of the above objective function is equivalent to the number of solutions of this equation. Since the LHS of this equation is a very simple function, it is apparent that the number of solutions to this equation is equivalent to the sign of the LHS evaluated at the single zero-derivative positive $q$. The derivative is $2\lambda_{W_{L-1}} L^2 q^{L-1} - (L-1)q^{L-2}$ which after solving for $q > 0$ yields $q = \frac{L-1}{2\lambda_{W_{L-1}} L^2}$. Plugging this back into the LHS above, we get the expression

$$2^{1-L} L^{1-2L} \lambda_{W_{L-1}}^{1-L} (L-1)^L - (L-1)^{L-1} 2^{1-L} \lambda_{W_{L-1}}^{1-L} L^{2-2L} + \frac{4L \lambda_{H_1} \lambda_{W_L} \prod_{j=1}^{L-2} \lambda_{W_j}}{\lambda_{W_{L-1}}^{L-2}}.$$

Thresholding this for 0 and simplifying, we get the final

$$\lambda_{H_1} \prod_{j=1}^{L} \lambda_{W_j} = \frac{(L-1)^{L-1}}{2^{L+1} L^{2L}}.$$

If the LHS is smaller than this threshold, the function in (18) has a single non-zero global optimum. If the LHS is at the threshold, the optimization problem has both 0 and some single non-zero point for a global solution. If the LHS is above the threshold, the optimization problem has only 0 as a single solution. $\qquad\square$

With all this, Theorem 3 follows very easily. We only need to conclude that if the optimization problem in Lemma 11 has a single non-zero solution, then both the optimal $s_{l,1}^*$ and $s_{l,2}^*$ are forced to be the same for each $l$. From this, we obtain the orthogonality of all the $\sigma(H_l^*)$ for $l \geq 2$. However, from Lemma 6 we know that even the optimal $H_l^*$ for $l \geq 3$ must be orthogonal, since it is non-negative and equal to $\sigma(H_l^*)$. From the same lemma, we conclude the statement about $W_l^*$ for $l \geq 2$. The optimal $W_1^*, H_1^*$ follow easily from Lemma 8.

By now, we only considered the case $n = 1$. Now, we proceed to the general $n$ case, where we need to also prove DNC1. We will proceed by contradiction. Assume there exist $\tilde{H}_1^*, W_1^*, \ldots, W_L^*$ which optimize $L$-DUFM, but $\tilde{H}_1^*$ is *not* DNC1-collapsed. Now, since the objective function in 10 is separable in the columns of $H_1$, we know

$$\frac{1}{2N} \left\| W_L^* \sigma(\ldots W_2^* \sigma(W_1^* h_{c,i}^{(1)})) - e_c \right\|_F^2 + \sum_{l=1}^{L} \frac{\lambda_{W_l}}{2} \|W_l^*\|_F^2 + \frac{\lambda_{H_1}}{2} \left\| h_{c,i}^{(1)} \right\|_2^2$$

$$= \frac{1}{2N} \left\| W_L^* \sigma(\ldots W_2^* \sigma(W_1^* h_{c,j}^{(1)})) - e_c \right\|_F^2 + \sum_{l=1}^{L} \frac{\lambda_{W_l}}{2} \|W_l^*\|_F^2 + \frac{\lambda_{H_1}}{2} \left\| h_{c,j}^{(1)} \right\|_2^2,$$

where $h_{c,i}^{(1)}, h_{c,j}^{(1)}$ are columns of $\tilde{H}_1^*$, for all $c, i, j$ applicable and $e_c$ is the $c$-th standard basis vector. Otherwise, we could have just exchanged all sub-optimal $h_{c,i}^{(1)}$ with the optimal one and achieve better objective value. However, if the last equality holds, then we can construct an alternative $\bar{H}_1^*$ where for each class $c$, we pick any $h_{c,i}^{(1)}$ and place it instead of all other $h_{c,j}^{(1)}$ within the class $c$. In this way, the $\bar{H}_1^*$ will be DNC1-collapsed, while still an optimal solution. However, for $\bar{H}_1^*$, we now know that if $H_1^*$ is constructed by taking just a single sample from both classes (thus forcing $n = 1$), this must be an optimal solution of the $n = 1$ $L$-DUFM with $n\lambda_{H_1}$ as the regularization term for $H_1$ and same feature vector dimensions and other regularization terms. This is because if there was a better solution for the corresponding problem, then we could construct a counterexample to optimality in the original $L$-DUFM problem by just simply taking this alternative, better solution and using the feature vectors of this solution for each sample of each individual class.

Therefore, since $H_1^*$ must be optimal in the new $n = 1$ problem (with a slight abuse of notation, we re-label the $n\lambda_{H_1}$ back to $\lambda_{H_1}$), we know that $\sigma(H_2^*)$ is collapsed in the way described by the theorem statement for $n = 1$. By non-negativity of $\sigma(H_2^*)$, this means that $\sigma(H_2^*)$ has two columns, let's call them $x, y$ for which: $x, y \geq 0; x^T y = 0, \|x\| = \|y\|$ (by the notation $x \geq 0$ we mean entry-wise inequality). This also implies that there is no such $i$ that $x_i, y_i > 0$. Moreover, let us assume $\sigma(H_2^*) \neq 0$. We can do that, as otherwise simple arguments would lead us into the degenerate solution where all $W_l^* = 0$, for which obviously all the columns in the originally optimal $\tilde{H}_1^*$ would need to be 0 implying simple DNC1 and reducing easily to the case of too big regularization explicitly stated in Theorem 3. We can now omit this case and assume non-triviality of the solution. We will first show that for a given set of optimal weight matrices $(W_1^*, W_2^*, \ldots, W_L^*)$, the matrix $\sigma(H_2^*)$ is uniquely determined regardless of what $H_1^*$ is. First, by the uniqueness of the solutions in Lemma 11 and by the fact that $s_{2i}$ depends monotonically on $\|x\|$, it follows that for any candidate direction $z, \|z\| = 1$, there is a unique multiplier $\alpha > 0$ for which $\alpha z$ can be a column of $\sigma(H_2^*)$, i.e. there is a uniquely determined norm under which any candidate direction $z$ must enter $\sigma(H_2^*)$. Now, however, by DNC3 for $l = 2$ we know that rows of $W_2$ are either zero or aligned with the columns of $\sigma(H_2^*)$. Moreover, since $H_3^*$ (which in case $L = 2$ shall be considered the output of the network) is orthogonal and thus rank 2, the weight matrix $W_2^*$ must contain both the columns of $\sigma(H_2^*)$ as its rows in direction. However, it cannot contain more than 2 different directions, otherwise for any

fixed optimal $\sigma(H_2^*)$, there would be an extra direction which would contradict DNC3. Thus, $W_2^*$ contains exactly two linearly independent (orthogonal) directions. But then, in turn, only these two directions can be columns of $\sigma(H_2^*)$. So $\sigma(H_2^*)$ is unique, possibly up to a switching of its columns $x, y$. However, only one permutation of these columns is optimal, because we know that the output of the matrix, $H_{L+1}$ is orthogonal and only one of its permutations is better aligned with $I_2$. Therefore $\sigma(H_2^*)$ is fully unique.

Invoking Lemma 9, we know that the whole range of matrices $A_2$ are such that $\sigma(A_2) = \sigma(H_2^*)$, while $\|A_2\|_* = \|H_2^*\|_*$. Namely, by Lemma 9, all the matrices $A_2$ which satisfy that condition are represented by having columns of the form $x - ty, y - tx$ for $t \in [0, 1]$. This can be derived as follows. According to the lemma and the description of $\sigma(H_2^*)$, this matrix is a good candidate for the non-negative part of the optimal solution of the corresponding optimization problem, and so the optimal solution (in the case of $L = 2$ a solution for which the nuclear norms of it and its non-negative part are equal) whose non-negative part equals $\sigma(H_2^*)$ exists. The negative values, according to Lemma 9, must follow a strict, yet still non-unique pattern. The negative values of the $(+, 0)$ rows of $\sigma(H_2^*)$ must be arranged so that they are aligned with the positive values in the corresponding rows, thus being multiple of $x$. The negative values of the $(0, +)$ rows must be, on the other hand, aligned with $y$. The coefficients of homothety between $x$ and the negative part in the $(+, 0)$ rows and between $y$ and the negative part of the $(0, +)$ rows must be equal due to the condition on equal $l_2$ norms of the negative parts corresponding to $(+, 0), (0, +)$ rows and the fact that $\|x\| = \|y\|$. Finally, the coefficients of homothety must be non-positive (yielding non-negative $t$) to retain the negativeness and no smaller than $-1$, due to the final condition of Lemma 9, stating that the product of any two negative values of counterfactually signed rows must be at most as big as the product of the corresponding positive entries and due to the already established alignment of positive and negative parts. In this way, the only matrices $A_2$ which satisfy these conditions are represented by having columns of the form $x - ty, y - tx$ for $t \in [0, 1]$, as stated above (note that $-ty$ is aligned with $x$ on $(+, 0)$ rows of $\sigma(H_2^*)$ and similarly for $-tx$).

Let us denote by $A_2^t$ a matrix of that form for a particular $t$. Now, the SVD of $A_2^t$ is as follows: $U^t$ is composed of $(x - y)/(\sqrt{2}\|x\|)$ and $(x + y)/(\sqrt{2}\|x\|)$, $\Sigma^t$ is a diagonal matrix with diagonal entries equal to $\|x\|(1 + t)$ and $\|x\|(1 - t)$, and $V^t$ is composed of the vectors $(1/\sqrt{2}, -1/\sqrt{2})^T$ and $(1/\sqrt{2}, 1/\sqrt{2})^T$. For $t \neq 0$, this is the unique SVD up to $\pm$ multiplications. For $t = 0$, $A_2^0$ is orthogonal and the SVD is non-unique. However, as we will see, this will not be relevant in the following argument. Invoking Lemma 8 we see that, for each $t$, the set of possible $W_1^t$ which can be the optimal solution is $\mathcal{S}^t = \{c_\lambda U^t (\Sigma^t)^{1/2} R^T\}$ for all possible orthogonal $R$ of suitable dimensions. From this representation, we see that the non-uniqueness of SVD is "wrapped" in the freedom of $R^T$. If $t = 0$, $\Sigma^0$ is diagonal and commutative and $U^0 R^T$ obviously cover all possible representations of $U$ in the SVD of $A_2^0$. If $t > 0$, we can easily cover the freedom in signs by $R$ too. Thus, we can assume a fixed $U$ and $V$ for all $t \in [0, 1]$. The question is, whether for $0 \leq t_1 < t_2 \leq 1$, $\mathcal{S}^{t_1} \cap \mathcal{S}^{t_2} = \emptyset$. To find out, consider some $R^{t_1}$ and $R^{t_2}$ orthogonal. We want to know whether it is possible that $c_\lambda U^{t_1}(\Sigma^{t_1})^{1/2}(R^{t_1})^T = c_\lambda U^{t_2}(\Sigma^{t_2})^{1/2}(R^{t_2})^T$. First, this obviously does not hold if $t_2 = 1$, because of the rank inequality. Assume $t_2 < 1$ and do the computation:

$$c_\lambda U^{t_1}(\Sigma^{t_1})^{1/2}(R^{t_1})^T = c_\lambda U^{t_2}(\Sigma^{t_2})^{1/2}(R^{t_2})^T \iff$$
$$U(\Sigma^{t_1})^{1/2}(R^{t_1})^T = U(\Sigma^{t_2})^{1/2}(R^{t_2})^T \iff$$
$$(\Sigma^{t_1})^{1/2}(R^{t_1})^T = (\Sigma^{t_2})^{1/2}(R^{t_2})^T \iff$$
$$(\Sigma^{t_1})^{1/2}(\Sigma^{t_2})^{-1/2} = (R^{t_2})^T(R^{t_1}),$$

where in the first equivalence we used the established knowledge that we can use the unique representative $U$ for all $t$. However, the diagonal elements of $(\Sigma^{t_1})^{1/2}(\Sigma^{t_2})^{-1/2}$ are $\sqrt{(1 + t_1)/(1 + t_2)}$ and $\sqrt{(1 - t_1)/(1 - t_2)}$. The first one is strictly greater than 1. On the other hand, all the entries of $(R^{t_2})^T(R^{t_1})$ are inner products between columns of orthogonal matrices and thus at most 1 in absolute value. This proves $\mathcal{S}^{t_1} \cap \mathcal{S}^{t_2} = \emptyset$ for $0 \leq t_1 < t_2 \leq 1$.

Thus, there is just a single $t^*$ for which $W_1^*$ is optimal and, moreover, there is a single $A_2^* = A_2^{t^*}$ that can be the output of the first layer. Then, if we analyze the two systems of linear equations $A_2^* = W_1^* X$, where $X$ is the $d \times 2$ matrix variable (each system for each column of $X$), we know that there is a unique min-$l_2$ norm solution per system. The min-$l_2$ norm solution must be the only optimal solution, because $H_1$ is regularized using the Frobenius norm. Thus, there is a single $H_1$

which can be the optimal solution. However, this is a contradiction with the assumption that we had a freedom while constructing $\bar{H}_1^*$ from $\tilde{H}_1^*$ – a possibility of choosing different columns from some classes as the representative of that class. Therefore, the $\tilde{H}_1^*$ must have had a single feature vector per class in the first place, which contradicts the assumption that it is not DNC1 collapsed.

Now that we have the DNC1 collapse, we can obtain the DNC2-3 collapse easily by just again going from the full $\tilde{H}^*$ to the reduced $n = 1$ version of it. Moreover, we know what are the forms of $W_1^*$ and $H_1^*$ – orthogonal transformations of the SVD of $H_2^*$. However, it must be said that these optimal matrices do not need to be orthogonal themselves, due to the $(\Sigma^t)^{1/2}$ multiplication, which makes the matrix non-orthogonal (consider for instance just taking the identity matrix extended to rectangularity for $R$). This concludes the proof of Theorem 3. $\qquad\square$

## B   Extension to multiple classes

Here, we comment shortly on why it is challenging to formulate an equivalent of Theorem 3 for $K > 2$. First of all, it is already unclear how to carry out the proof of Lemma 5 for $K > 2$ and even whether it holds. However, the bigger issue lies in Lemma 9. Note that $\|H_2\|_* \geq \|\sigma(H_2)\|_*$ only holds for matrices with two rows or columns. A simple counterexample with $3 \times 3$ matrix goes as follows:

$$ A = \begin{pmatrix} -1 & 0 & 1 \\ 0 & 1 & 1 \\ 1 & 1 & 0 \end{pmatrix}, $$

for which $\|A\|_* = 3.464$ and $\|\sigma(A)\|_* = 3.494$. However, this does not mean that Theorem 3 does not hold for $K > 2$. For instance, at orthogonal matrices, which are collapsed, one can show that this equation holds by computing the sub-gradient of the nuclear norm and using convexity. It is also hard to come up with counterexamples to the inequality for matrices which are close to orthogonal. Therefore, it is our belief that the cost of losing orthogonality in other loss terms makes the compensation of achieving slightly lower nuclear norm insufficient. However, to make this argument formal, one needs a sophisticated control over this quantity together with all the other results.

## C   Further numerical experiments on DUFM and ablations

In this section, we further investigate the effect of significant parameters on the training dynamics of DUFM. We focus on the effect of depth, regularization strength and width on the outcome of DUFM optimization. All the plots (except the one in Appendix C.5) are averaged over 10 runs with one standard deviation to both sides displayed as a confidence band. First, however, we supplement the numerical results from Section 4.2 with the promised plots of the losses.

### C.1   Extra material to the numerical results in Section 4.2

In Figure 3, we plot the loss functions and theoretical optimum, together with the disentangled loss functions computed separately on the fit term and the regularization terms for $H_1$, $W_l$, with $1 \leq l \leq L$. As already discussed in Section 4.2, the DUFM finds the global optimum perfectly and exhibits superb DNC.

### C.2   Effect of depth on DUFM training

As already seen by showing 3- and 6-layered DUFM, the depth does not play a significant role on the training of DUFM. However, an implicit bias of GD can be observed for very deep DUFMs. To demonstrate this, we plot the 10-DUFM training results in Figure 4.

Though not well visible on the plots, the DNC2 is *sometimes* achieved also on $H_1, H_2$, which in the shallower DUFMs was an exception. This shows that with increasing depth, GD is more and more implicitly biased towards finding DNC2 representations for these feature matrices too.

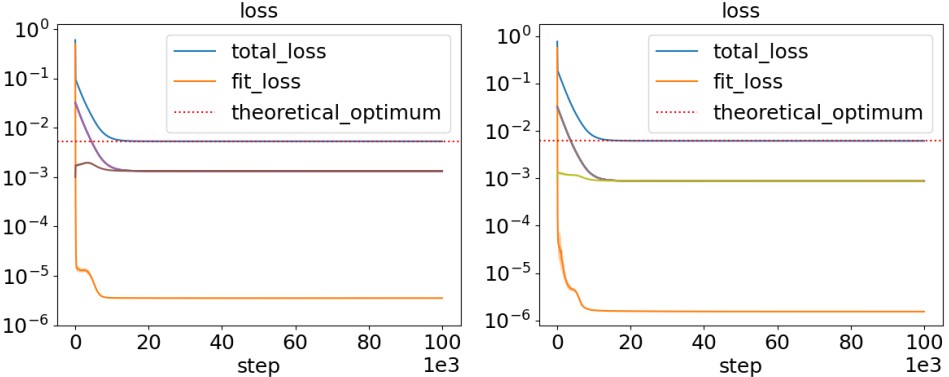

Figure 3: Losses and partial losses for 3-DUFM (left) and 6-DUFM (right) as described in Section 4.2. The total training loss is compared to the numerically computed optimum (based on (17)). The unlabeled loss curves correspond to all the regularization losses.

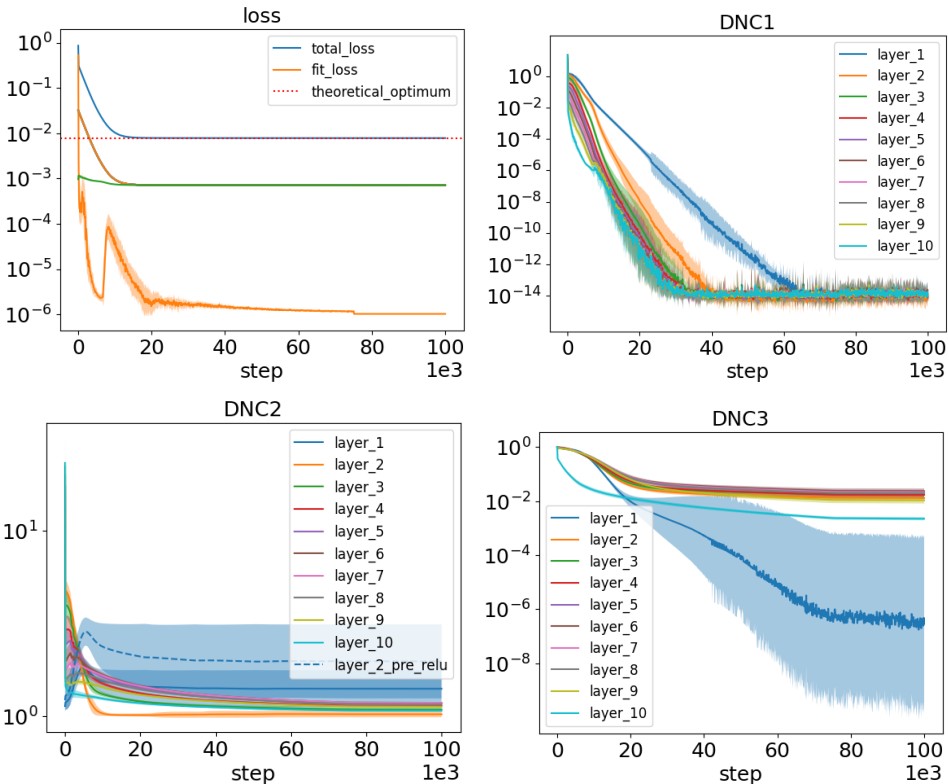

Figure 4: Losses and DNC metric progressions as a function of training time. The unlabeled loss curves correspond to all the regularization losses. Here, the 10-DUFM is trained. The variance in some of the DNC metrics is due to the non-zero chance of qualitatively different behaviors for the first and second layer.

### C.3 Effect of width on DUFM training

Unlike depth, width has a significant effect on the DUFM training, especially at small widths. The reason is that the model parameters often get stuck in sub-optimal configurations due to the difficult optimization landscape. To compare, we ran 4-DUFM training with the same hyperparameters except the width. We repeated the experiments for widths of 2, 5, 10, 20, 64, 1000. We plot all the results in Figure 5.

The results suggest that the risk of DUFM not reaching the global optimum and thus DNC drastically decreases with the width. Already for width 20, the DUFM did not fail to reach the global optimum once in ten runs. On the other hand, for width of only 2, the DUFM did not reach the global optimum in any run. Interestingly, the smaller widths yielded less expected behaviour. For instance for width 10, we encountered two runs both reaching a perfect optimum and DNC, of which one exhibited DNC2 also for $H_1, H_2$, while the other did not. Noticeably, for width 1000 we see higher variance in some DNC metrics. This is because some of these specific metrics converged to very low values for some runs and for some other stayed comparably higher. This effect appears to be an implicit bias of SGD.

### C.4 The effect of weight decay on DUFM training

As expected for the MSE loss setup, the weight decay is the main driving force for the occurrence of DNC. Therefore, predictably, the weight decay has a direct influence on the speed of convergence towards both DNC and global optimality. This is clearly demonstrated in Figure 6, where we plot the training dynamics of 4-DUFM with width 64 and weight decays of sizes $0.0001, 0.0005, 0.0009, 0.0013, 0.0017$. By increasing the value of the weight decay, the loss as well as DNC metrics converge much faster. However, there is no qualitative change in the result of the training for different values of weight decay.

### C.5 Interconnection of DNC and optimality of DUFM

As an interesting bonus, we show a single, cherry-picked run for the 4-DUFM with width 10 (although such runs occur quite often at this width). This training first got stuck at an almost-optimal solution, then after a long stay in the saddle point, it jumped to the optimum. The DNC metrics showed a clear decrease at exactly that point of the training, as shown in Figure 7. This run clearly shows how interconnected the deep neural collapse is with optimality.

## D    Pre-trained DUFM experiments on CIFAR-10

Here, we specify the training conditions from Section 4.2 in more detail, and add results of further runs to provide more robust evidence. We use three different architectures - ResNet20, DenseNet121 and VGG13. We only impose one change to these - the standard backbone is extended with one extra fully-connected layer with output dimension 64 to enable free features and reasonable size of the DUFM. The weights are initialized with He initialization [9] with scaling 1. The DUFMs in question are 3- and 5-layered of constant width 64. We train with weight decay 0.00001 and with learning rate 0.001 on 1000000 full GD steps. The backbone was then trained using stochastic gradient descent with batch size 128, learning rate 0.00001 and with 10000 epochs on classes 0 and 1 of CIFAR-10. While in Figure 1 we plot results of a single training of the 3-DUFM model followed by 7 independend trainings of independent initializations of the ResNet20, here we add two more independent trainings of the DUFM averaged over further 7 independent runs of ResNet20 training. The results are qualitatively the same as in the main body, which suggests that the ability of ResNet20 to make unconstrained features suitable for the emergence of DNC is robust w.r.t. the well-trained DUFM head. The results of these two runs are depicted in Figure 8.

As a side note, we remark that unlike end-to-end training or DUFM training, the ResNet20 training with pre-trained DUFM head is more prone to undesired behaviours. First of all, we need a rather small learning rate to prevent divergence and even with that, a minority of runs never jumps from $50\%$ training accuracy. Therefore, in our results, we only report those runs, which converged to near $100\%$ training accuracy.

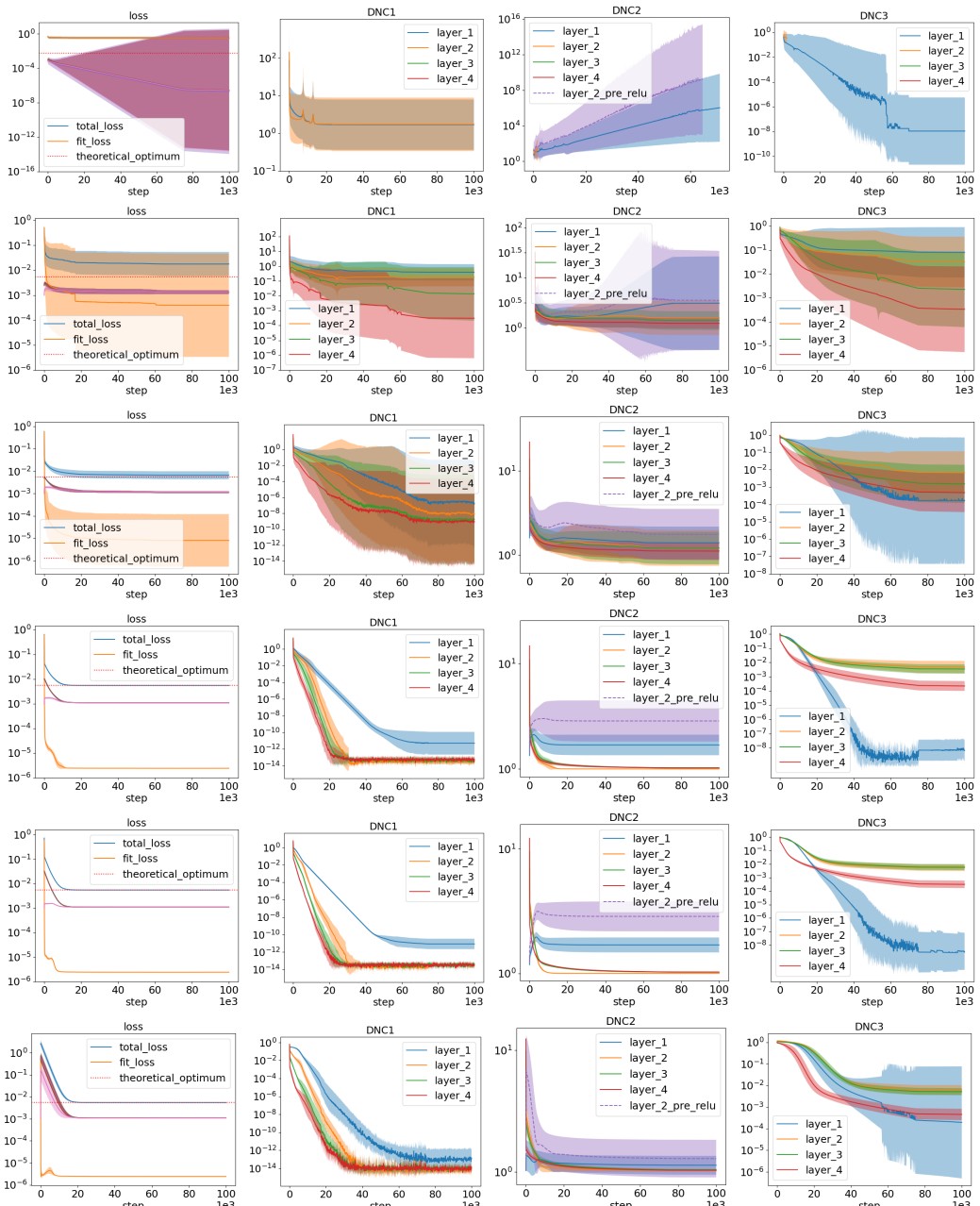

Figure 5: Losses and DNC metric progressions as a function of training time. The unlabeled loss curves correspond to all the regularization losses. Each row represents a different width, in increasing order ranging through $\{2, 5, 10, 20, 64, 1000\}$. As the width increases, the training gets more stable. For width 2, no training converges to optimum. For width 5, the training only rarely converges to optimum. For width 10, the training usually converges to optimum, but often after encountering saddle points. From width 20, we see a stable training behavior with almost no difference between different high-width trainings.

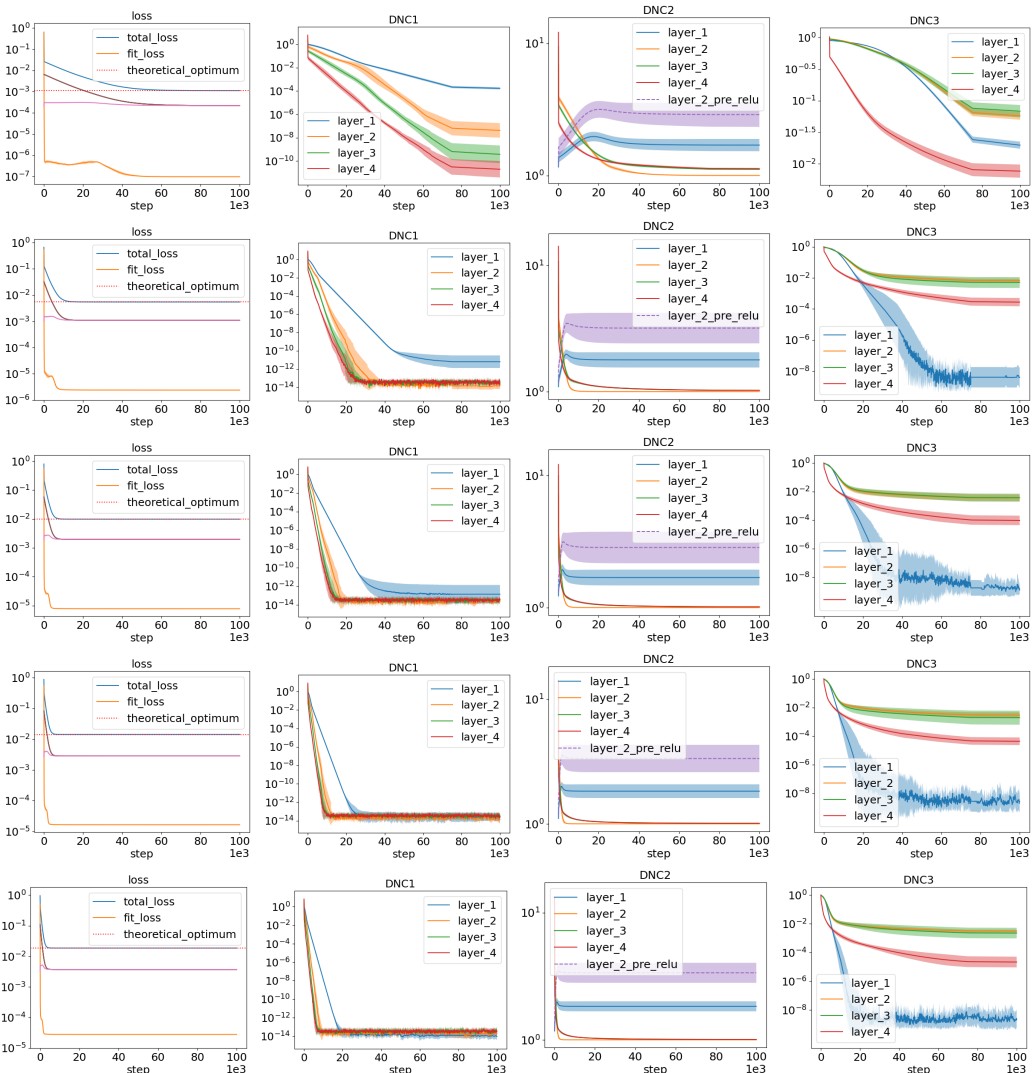

Figure 6: Losses and DNC metric progressions as a function of training time. The unlabeled loss curves correspond to all the regularization losses. Each row represents a different value of weight decay in increasing order through a set $\{0.0001, 0.0005, 0.0009, 0.0013, 0.0017\}$. By increasing the weight decay, the training dynamics get faster.

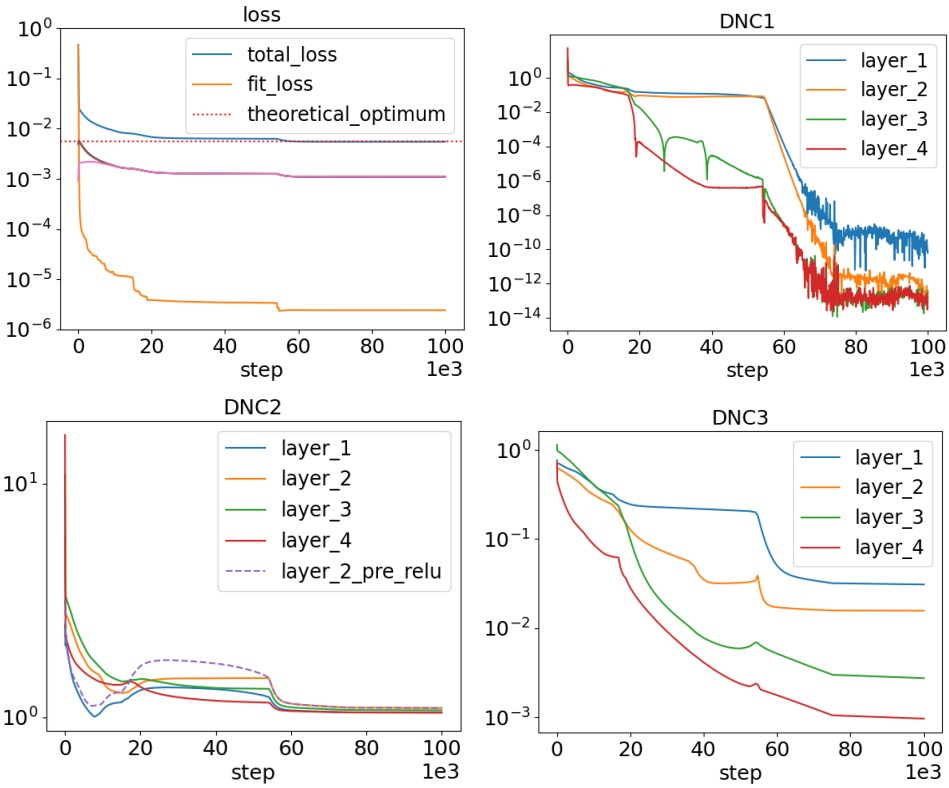

Figure 7: Losses and DNC metric progressions as a function of training time. The unlabeled loss curves correspond to all the regularization losses. The plot shows a single run of the training of 4-DUFM with width 10. After staying in a close-to-optimal saddle point, the model achieved optimum, the DNC metrics rapidly decreasing.

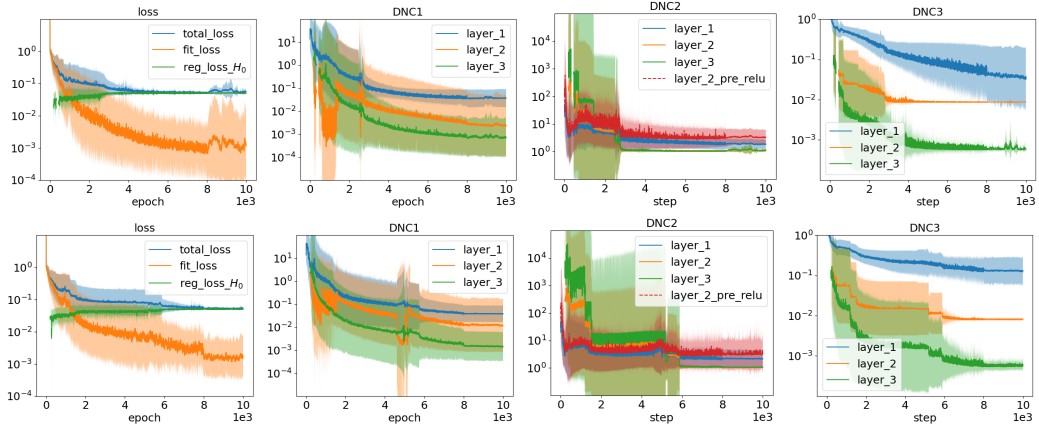

Figure 8: Loss functions and deep neural collapse metrics as a function of training progression. The rows correspond to different pre-training runs of the 3-DUFM model. On the left, the training loss function and its decomposition are displayed. Next, the DNC1-3 metrics are displayed. The ResNet20 recovers the DNC metrics for DUFM, in accordance with our theory. This validates unconstrained features as a modeling principle for neural collapse.

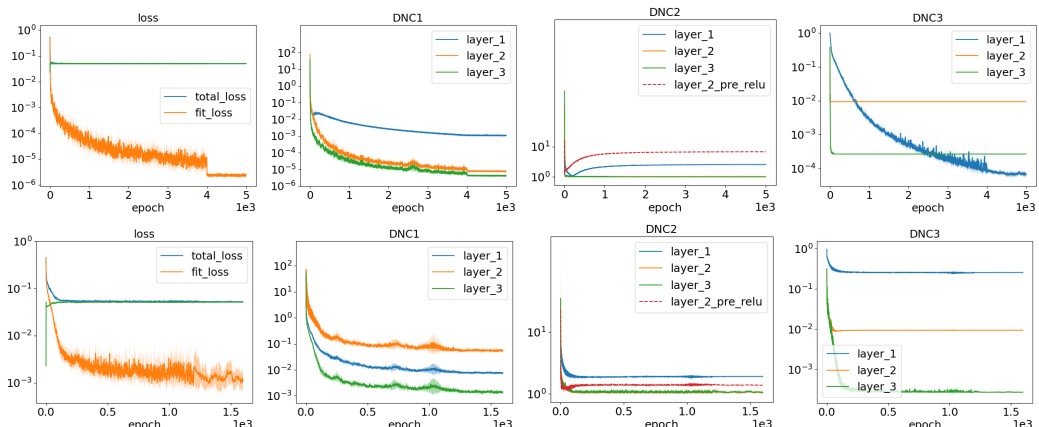

Figure 9: Loss functions and deep neural collapse metrics as a function of training progression. The first row corresponds to VGG13 experiments, the second row corresponds to DenseNet121 experiments with the pre-trained 3-DUFM model. On the left, the training loss function and its decomposition are displayed. Next, the DNC1-3 metrics are displayed. Both architectures recover the DNC metrics for DUFM, in accordance with our theory. This validates unconstrained features as a modeling principle for neural collapse.

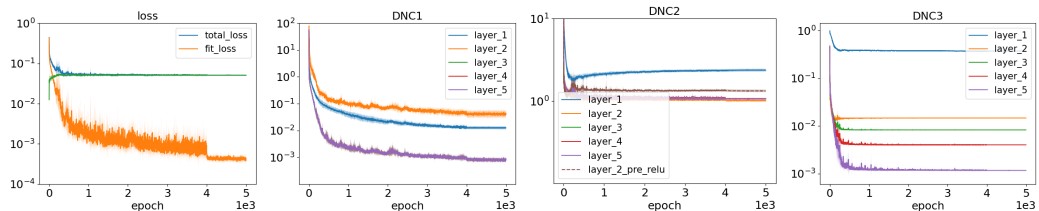

Figure 10: Loss functions and deep neural collapse metrics as a function of training progression. The head considered is a pre-trained 5-DUFM model. On the left, the training loss function and its decomposition are displayed. Next, the DNC1-3 metrics are displayed. The ResNet20 recovers the DNC metrics for DUFM, in accordance with our theory. This validates unconstrained features as a modeling principle for neural collapse.

We further present the results of the same experiment but with 5-DUFM attached to ResNet20 as well as 3-DUFM attached to DenseNet121 and VGG13, to even further validate the robustness of our results. These results are presented in Figures 9 and 10, and they are in full accordance with the ResNet20 experiments.

