# OpenReview forum: "Deep Neural Collapse Is Provably Optimal for the Deep Unconstrained Features Model"
_NeurIPS.cc/2023/Conference — NeurIPS 2023 spotlight_

### Official Review · Reviewer_FUzi · 2023-07-02

**Soundness:** 4 excellent
**Presentation:** 4 excellent
**Contribution:** 3 good
**Rating:** 7
**Confidence:** 4

**Summary:**

The paper introduces an extension to the standard unconstrained features model in neural collapse theory by incorporating multiple layers above the unconstrained features. The study demonstrates that, under certain conditions and at optimality, all of the top layers exhibit all properties of neural collapse.

**Strengths:**

-- The paper exhibits excellent writing and effectively surveys the existing literature.
-- The paper contributes to a promising direction in the theory of neural collapse by presenting a significant extension, and it conducts a non-trivial theoretical analysis that offers compelling evidence for the occurrence of neural collapse at intermediate layers. This extension aligns with previous empirical findings, verifying the theoretical predictions made in earlier studies.



**Weaknesses:**

-- Like other papers in the field, the rationale behind asserting that SGD converges to a solution satisfying neural collapse solely based on optimality is not entirely clear. It would greatly benefit the paper if the authors could provide further clarification and expand upon this aspect.
-- While the paper focuses on the binary classification case, which is acceptable, it would be valuable to discuss the feasibility and potential challenges of extending the analysis to a multiclass setting.
-- I believe that expanding on the implications and interpretations of the results would significantly improve the paper. It would be beneficial for the authors to discuss how these findings relate to aspects like generalization and what implications they have for our understanding of deep learning in general.

**Questions:**

See weaknesses.

**Limitations:**

There are no negative societal issues with this paper.

---

> ### Author Rebuttal · Authors · 2023-08-09
>
> We thank the reviewer for the insightful review and positive feedback. We address all the points raised in the review below.
>
> **The rationale behind asserting that SGD converges to a solution satisfying neural collapse solely based on optimality is not entirely clear**
>
> *Response.* We fully agree with the reviewer that the ‘static’ global optimality analysis by itself does not prove SGD convergence. Showing SGD convergence (and, thus, providing a ‘dynamic’ analysis of DNC) is a challenging future research direction.
>
> We note that the full SGD convergence is not completely established even for the standard neural collapse, with [7, 21] giving partial solutions based on the assumptions discussed in our related work section.
>
> [12, 20, 42] try to argue the convergence by showing that the loss landscape of the unconstrained features model is benign. An equivalent statement for the DUFM is unlikely to even be true. In fact, [R1] shows that multi-layer deep linear networks exhibit spurious local minima. Including the ReLU activation intuitively does not simplify the loss landscape, nor does it make it benign.
>
> **While the paper focuses on the binary classification case, which is acceptable, it would be valuable to discuss the feasibility and potential challenges of extending the analysis to a multiclass setting**
>
> *Response.* This issue was raised by multiple reviewers. Hence, we have opted to respond to it in the global response above.
>
> **It would be beneficial to discuss the practical implications of our findings and how they relate to topics such as generalization and general understanding of neural networks.**
>
> *Response.* This is a good suggestion. We provide below three insights resulting from our theory.
>
> *(i)* Our results suggest that the $l_2$ regularization leads the deep neural network to find solutions with well-conditioned matrices (orthogonal in the ideal DNC case). This in turn provides more robust models, which also suggests better generalization.
>
> *(ii)* In principle, it should be possible to exchange the $l_2$ regularization on the weight matrices of the backbone with an $l_2$ regularization directly on $H_1$. In this way, we expect to be able to force the occurrence of the DNC more directly. Some works [1, 5, 6, 15, 16, 17, 32, 34] try to utilize the neural collapse for practical purposes (including transfer learning, out-of-distribution detection, robustness, and calibration) and this is a good way to better reach it.
>
> *(iii)* As for the general understanding of neural networks, we can learn from the DNC what is the ideal state at which deep neural networks are incentivized to represent the data, having been trained with MSE loss and $l_2$ regularization. In particular, the network tends to reduce the within-class variability and create structured feature representations. This suggests that we should treat the terms “features” and “feature extraction” with caution, as two different samples from the same class will have the same feature representation, despite being significantly different in nature (such as two images of a dog with considerably different background or coming from considerably different dog species). The after-relu DNC2 results of Theorem 3 suggest that two samples of different classes, having orthogonal, yet non-negative feature representations, cannot share the activation at any of the neurons. This means that even seemingly similar samples with different labels do not share the same feature representation at any neuron, despite perhaps sharing some major features (such as two different species of a dog with a different label, sharing most of the dog’s visual properties and only differing in fine details).
>
> [R1] Kawaguchi, Kenji. "Deep learning without poor local minima." Advances in neural information processing systems 29 (2016).

---

### Official Review · Reviewer_Egs3 · 2023-07-04

**Soundness:** 3 good
**Presentation:** 4 excellent
**Contribution:** 3 good
**Rating:** 7
**Confidence:** 4

**Summary:**

The paper studied Deep Neural Collapse (DNC) which extends the structure of Neural Collapse of the last layer to multi-layers in deep neural networks. Theoretically, this paper generalizes the established unconstrained features model to the deep unconstrained features model (DUFM) of multi-layer non-linear models for binary classification. Based on the deep unconstrained features model, they show the global optimum exhibits several extended properties. Empirical experiments demonstrate that the results of the optimized deep unconstrained features model and trained networks agree with the theory.


**Strengths:**

1. The paper generalizes the Neural Collapse to Deep Neural Collapse which describes the structure of the feature other than the last layer. The proof based on the deep unconstrained features model for binary classification is clear and easy to follow. It gives a framework to find and prove the feature structure of other layers in DUFM.

2. The extensive experiments and numerical results shows that the theory successfully predicts the behavior of the deep unconstrained features model(DUFM) which exhibits proposed DNC properties.

3. The paper is well written, it is smooth to follow the whole logic and the idea of the method.


**Weaknesses:**

1. Though the model and proof consider the Deep Neural Collapse which works for not only features of the last layer, the model and the proof are restrictive which works for binary classification and no bias case. Many proofs and derivations rely on binary classification like Lemma 5 - Lemma 7 which is the key to giving the conditional optimum of previous layers.

2. In addition, the DUFM adds a regularization term for the first layer feature which is not fit with the real training of neural networks to some extent.

3. Compared to the numerical results of DUFM, the experiments for real training of neural networks are limited to the 3-layer Resnet20. If there are more results for the training of other neural networks it will make the experiments more convincible.

4. There are some typos in the proof, like L517 in the appendix where I think the $\sigma_1^2$ should be $s_{L, i}^2$.

**Questions:**


1. Most previous research[1,2, 3] focus on models with bias while in [3] they also study the unconstrained features model without bias and show the features exhibit OF rather than STF under UFM with bias. So I am just wondering if it is possible to extend the DUFM to the bias case and show the STF structure in DNC2?

2. There are some experiments results for 3-layer Resnet-20 in the paper which also show DNC, but in [4] the results seem to show that the Neural Collapse process gradually across the layers of the neural network like the measure of NC1 decreases exponentially. I think this may be caused by the gap between the DUFM and real trained networks. Are there more experiments on real trained networks and whether they exhibit DNCs? If so, why is there such a difference between the results in this paper and the results in [4]. If not, is that mean there is a gap and needs other explanations for such a phenomenon?

3. All experiments based on DUFM in this paper add the regularization term of the first layer feature. By the derivation of the DUFM, it seems that possible to add a regularization term on the intermediate layer feature, for example, add $||H_5||_2$ and optimize $W_l, l \ge 5$, are there any experiment results can demonstrate in this case how the DNC performs?


Reference:
[1] Zhu, Z., Ding, T., Zhou, J., Li, X., You, C., Sulam, J., & Qu, Q. (2021). A geometric analysis of neural collapse with unconstrained features. Advances in Neural Information Processing Systems, 34, 29820-29834.

[2] Fang, C., He, H., Long, Q., & Su, W. J. (2021). Exploring deep neural networks via layer-peeled model: Minority collapse in imbalanced training. Proceedings of the National Academy of Sciences, 118(43), e2103091118.

[3] Tirer, T., & Bruna, J. (2022, June). Extended unconstrained features model for exploring deep neural collapse. In International Conference on Machine Learning (pp. 21478-21505). PMLR.

[4]He, H., & Su, W. J. (2022). A law of data separation in deep learning. arXiv preprint arXiv:2210.17020.

**Limitations:**

The framework and the proof relies on the binary classification and no bias case which make the DUFM not so correlated to the real trained networks.

---

> ### Author Rebuttal · Authors · 2023-08-09
>
> We thank the reviewer for the positive, insightful and engaging review, and for pointing out the strengths of our work. We address all the points raised in the review below.
>
> **The paper only considers the binary classification and the bias-free case**
>
> *Response.* For the extension to multiple classes, please see the global response. As for the omission of the bias term, we believe it is possible to extend our analysis to include the bias. In fact, Lemmas 5-9 are unchanged, and the only changes involve Lemma 4 and Lemma 11, which can also be adjusted to accommodate the bias. However, by the argument made in [29] that the ETF structure can be obtained from the orthogonal frame (OF) structure by simply centering it, we believe that including bias would not provide significant new insights on top of our existing results. Since the computations would get much more complicated, we thus opted to omit this case.
>
> **DUFM adds a regularization term for the first layer feature which is not fit with real training of neural networks**
>
> *Response.* We agree with this point. Indeed, the regularization can be regarded as a simplification with respect to the (more complicated) nature of real training in which features are the output of regularized network layers. However, $l_2$ regularization on the unconstrained features is common to most works considering regularized UFM, see e.g. [2, 7, 25, 27, 28, 29, 30, 40, 42].
>
> For standard last-layer neural collapse, global optimality was also demonstrated under different assumptions, such as $l_2$-norm constraints [19, 31, 36]. Thus, an interesting direction for future work is to consider constrained features instead of the $l_2$ regularization, in the context of the deep neural collapse.
>
> Finally, we highlight that our results can be readily generalized to the case where $H_1$ is regularized with any Schatten-$2/L$ pseudonorm. This pseudonorm constitutes a candidate for a more accurate regularization term. In fact, [R1] shows that regularizing a sequence of weight matrices without the ReLU activation (i.e., in a deep linear network) is equivalent to regularizing their product with the Schatten-$2/L$ pseudonorm, where $L$  is the number of the weight matrices.
>
> **Experiments for real training of neural networks are limited to the Resnet20 with 3-layer MLP on top.**
>
> *Response.* In the PDF attached to the general response above, we provide additional experiments:
>
> *(i)* We consider VGG13 and DenseNet121 architectures, keeping the rest of the experimental setup the same as in Figure 1 of the paper.
>
> (ii) We consider a 5-layer head on top of the ResNet20.
>
> All these new experiments fully agree with the original one and support our theory.
>
>
> **Q1: is it possible to extend the DUFM to the bias case and show the STF structure in DNC2?**
>
> *Response.* We believe it is possible to do our analysis with the bias term included and this would yield results similar to the one-layer NC. Please see our response above in the weaknesses section.
>
> **Q2: In [4] the results seem to show that the Neural Collapse progresses gradually across the layers of the neural network like the measure of NC1 decreases exponentially**
>
> *Response.* We agree that there is a gap between the predictions of the DUFM and the progressive neural collapse which has been observed empirically in real networks, and investigating this gap is an important future direction. Let us make two comments in this regard.
>
> *(i)* The progressivity of DNC may be due to the non-linearity of the data, which the neural network has to eliminate throughout the layers. One reason for the gradual removal of within-class variability through the layers could be the balance between the norms of the weight matrices of different layers. In fact, decreasing the within-class variability more quickly would require larger matrix norms, which is prevented by the regularization.
>
> *(ii)* A dynamic analysis of the trajectory of gradient descent could also show a progressive neural collapse: the final layers reduce the within-class variability faster than the first ones, as training progresses.
>
> Finally, we note that the phenomenon of progressive neural collapse is not entirely understood even at the experimental level: on the one hand, [8] observes a log-linear behavior in the DNC1 reduction; on the other hand, [26] observes a reduction which is no longer log-linear.
>
> **Q3: Is it possible to directly regularize the intermediate features instead of the weight matrices from previous layers?**
>
> *Response.* Yes, this is indeed possible. This would also provide a more direct way of enforcing the occurrence of DNC, which could be beneficial for the applications of neural collapse (including e.g. transfer learning, out-of-distribution detection, robustness, and calibration, see [1, 5, 6, 15, 16, 17, 32, 34]).
>
> [R1] Shang, Fanhua, Yuanyuan Liu, and James Cheng. "Unified scalable equivalent formulations for Schatten quasi-norms." arXiv preprint arXiv:1606.00668 (2016).

---

> > ### Comment · Reviewer_Egs3 · 2023-08-13
> >
> > Thanks for the detailed response and additional experiments. I think most of the questions are well answered and further experimental results fit with the theory. So I change my Rating from 6 to 7.

---

### Official Review · Reviewer_9Ryi · 2023-07-05

**Soundness:** 3 good
**Presentation:** 3 good
**Contribution:** 3 good
**Rating:** 6
**Confidence:** 4

**Summary:**

The paper proposes a novel approach to investigating deep neural collapse in deep neural networks by presenting a generalization of the analytical framework for neural collapse (NC) to multiple non-linear layers. The paper introduces the deep unconstrained features model to demonstrate that the unique global optimum of the MSE loss for binary classification exhibits all the properties typical of deep neural collapse. However, the paper only considers the MSE loss without a bias term for binary classification, which may limit the applicability of deep neural collapse for other losses. Specifically, the theoretical conclusions may only be provably optimal in some cases.

**Strengths:**

- The paper presents a  generalization of the unconstrained features model  to multiple non-linear layers, filling a gap in the existing literature.
- The paper contributes to a better understanding of deep neural collapse in binary classifcation deep neural networks, which has important implications for the design and optimization of deep learning models.

**Weaknesses:**

- The theoretical analysis only involves binary classification, and it is unclear whether the proposed approach can be extended to multi-class classification problems, especially for a large number of classes that would change the final form (DNC2) of the resulting deep neural collapse.
- The paper only considers the MSE loss without a (last-layer) bias term, which may limit the applicability of deep neural collapse for other losses. Specifically, benifiting from the MSE loss without the bias term, the unique global solution can exhibit close-formed results, which may further allow the derivation of the proof to be greatly simplified.
-  while neural collapse induces a simplex equiangular tight frame [1], which may not hold for other loss functions  or if a bias term is included [2].
- The paper lacks extensive experiments to validate deep neural collapse. Additional experiments on a variety of datasets (especially for a large number of classes) and deep neural networks would help to better understand and confirm the prevalence of deep neural collapse.
- The paper could benefit from a more thorough discussion of the practical implications of the proposed approach for the design and optimization of deep learning models.

[1] Vardan Papyan, X. Y. Han, and David L Donoho. Prevalence of neural collapse during the terminal phase of deep learning training. In *Proceedings of the National Academy of Sciences (PNAS)*, volume 117, 2020.

[2] Jinxin Zhou, Xiao Li, Tianyu Ding, Chong You, Qing Qu, and Zhihui Zhu. On the optimization landscape of neural collapse under mse loss: Global optimality with unconstrained features. In *International Conference on Machine Learning (ICML)*, 2022.



**Questions:**

see weaknesses.

---

> ### Author Rebuttal · Authors · 2023-08-09
>
> We thank the reviewer for the positive and insightful assessment. We address all the points raised in the review below.
>
> **Extension to multi-class classification problems**
>
> *Response.* This issue was raised by multiple reviewers. Hence, we have opted to respond to it in the global response above.
>
> **The paper only analyzes the MSE loss without the bias term**
>
> *Response.* We agree that using the MSE loss does not provide a full insight into the usage of other losses. Indeed, as we see in the literature on neural collapse, the analysis and even the statements for MSE and cross-entropy (CE) losses differ significantly. For instance, the neural collapse for CE loss is not well-defined without feature normalization or weight decay.
>
> In our work, we perform the analysis with the MSE loss, as it provides more options for mathematical treatment. We highlight that the MSE loss is a reasonable choice, as it is reported by [R1] that training with the MSE loss can perform on par with the training with the CE loss. For these reasons it is the setup of choice of a large number of papers on neural collapse [7, 21, 28, 30, 40, 41]. An extension to cross-entropy is a great direction for future work, however the corresponding analysis is likely to be quite different. In fact, the CE loss does not admit closed-form solutions, and it typically yields a structure described by an equiangular tight frame (ETF) at least in the last layer.
>
> As for the omission of the bias term, we believe it is possible to extend our analysis to include the bias. In fact, Lemmas 5-9 are unchanged, and the only changes involve Lemma 4 and Lemma 11, which can also be adjusted to accommodate the bias. However, by the argument made in [29] that the ETF structure can be obtained from the orthogonal frame (OF) structure by simply centering it, we believe that including bias would not provide significant new insights on top of our existing results. Since the computations would get much more complicated, we thus opted to omit this case.
>
>
> **The paper lacks experiments to validate the neural collapse**
>
> *Response.* We agree that validating the DNC is a key prerequisite for its theoretical study. However, this validation was already extensively made in recent work (see [8, 26]). Our contribution crucially builds upon this experimental evidence and provides a theoretical analysis of the same phenomenon.
>
> That being said, we still do experiments related to main take-away of our paper:
>
> *(i)* We have performed a rather extensive set of experiments on DUFM: different depths (ranging from 2 to 10), different widths (from very small to as wide as 1000 neurons), different weight decays, and different learning rates. Many of these experiments are discussed in detail in the ablation study in Appendix C, where we provide a concise description of the effects these hyperparameters have on the emergence of DNC.
>
> *(ii)* The experiments on real data validate the DUFM assumption, a point that was lacking in the related literature, see both the main paper and the additional results in Appendix D. We also provide extra experiments in the 1-page PDF attached to the global response above:
>
> *(ii-a)* We consider VGG13 and DenseNet121 architectures, keeping the rest of the experimental setup the same as in Figure 1 of the paper.
>
> *(ii-b)* We consider a 5-layer head on top of the ResNet20.
>
> All these new experiments fully agree with the original one and support our theory.
>
> **The paper could benefit from further discussion on the practical applications of the proposed approach**
>
> *Response.* This is a good suggestion. We provide below two insights resulting from our theory.
>
> *(i)* Our results suggest that the $l_2$ regularization leads the deep neural network to find solutions with well-conditioned matrices (orthogonal in the ideal DNC case). This in turn provides more robust models.
>
> *(ii)* In principle, it should be possible to exchange the $l_2$ regularization on the weight matrices of the backbone with an $l_2$ regularization directly on $H_1$. In this way, we expect to be able to force the occurrence of the DNC more directly. Several works (e.g., [1, 5, 6, 15, 16, 17, 32, 34]) try to utilize the neural collapse for practical purposes (including transfer learning, out-of-distribution detection, robustness, and calibration), and this is a good way to better reach it.
>
> [R1] Demirkaya, Ahmet, Jiasi Chen, and Samet Oymak. "Exploring the role of loss functions in multiclass classification." 2020 54th annual conference on information sciences and systems (ciss). IEEE, 2020.

---

### Official Review · Reviewer_nhUP · 2023-07-10

**Soundness:** 3 good
**Presentation:** 3 good
**Contribution:** 2 fair
**Rating:** 6
**Confidence:** 5

**Summary:**

This paper introduces the concept of deep neural collapse (DNC), extending the understanding of neural collapse to earlier layers of deep neural networks. It proposes the deep unconstrained features model (DUFM) as a theoretical framework to analyze DNC. The authors demonstrate that for binary classification, DNC is a globally optimal solution in the DUFM, providing rigorous evidence of its occurrence in multiple layers. Numerical experiments confirm that gradient descent efficiently finds solutions in line with the theory, exhibiting neural collapse in multiple layers. The paper concludes by highlighting open problems for future research, such as generalizing the analysis to multiple classes, exploring gradient dynamics in the DUFM, and understanding the impact of biases and loss functions on the collapsed solutions.

**Strengths:**

$\bf (1)\ \textbf{Strong Technical Analysis}:$ A key strength of this paper is its strong and rigorous technical analysis. The authors provide a solid theoretical framework, specifically the deep unconstrained features model (DUFM), to analyze and understand the phenomenon of deep neural collapse (DNC). Theorems and empirical evidence are presented to support their findings, enhancing the credibility and reliability of the results.

$\bf (2)\ \textbf{Extension to Multiple Non-linear Layers:}$  This paper addresses a crucial gap in the existing literature by extending the analysis of neural collapse beyond the last layer to multiple non-linear layers. By generalizing the unconstrained features model, the authors provide insights into deep neural collapse (DNC) occurring in deeper layers of neural networks. This extension contributes to a more comprehensive understanding of the behavior of deep networks during training.

$\bf (3)\ \textbf{Empirical Validation:}$ The paper provides empirical evidence to validate the theoretical framework. The authors demonstrate that by optimizing deep unconstrained feature models using gradient descent, the resulting solutions align well with the theoretical predictions. This empirical validation reinforces the credibility of the proposed modeling principles and their applicability in real-world scenarios.

$\bf (4)\ \textbf{Relevance and Contribution to the Field:}$ The paper addresses a significant topic of interest in the field of deep learning by focusing on the phenomenon of neural collapse in deep neural networks. By providing theoretical insights and empirical evidence, the paper contributes to the existing body of knowledge and expands our understanding of the behavior of deep networks during training.

**Weaknesses:**


$\bf (1)\ \textbf{Theorem 3 cannot explain the progressive neural collapse:}$ In Ref [8], the authors observe that each layer roughly improves a certain measure of data separation by an equal multiplicative factor. A similar observation is depicted in Figure 1 of the current paper. However, Theorem 3 contradicts this observation by demonstrating that for all layers $l \ge 2$, the within-class variability of features becomes zero, and the features form an orthogonal matrix. This discrepancy suggests that there may be additional factors or dynamics at play in the behavior of the layers beyond what is captured by the initial observation. Further investigation is necessary to reconcile these contrasting findings and gain a more comprehensive understanding of the behavior and properties of features in deep neural networks.

$\bf (2) \textbf{Role of activations:}$ One weakness of the paper is the limited discussion or exploration of the specific role and impact of activations in the context of deep neural collapse (DNC). While the paper acknowledges that DNC1 and DNC2 hold for layer l ≥ 2 before and after activations, it does not provide an in-depth analysis or explanation of how the activations influence or interact with DNC.

$\bf (3) \textbf{Limited Scope:}$  The paper primarily focuses on deep neural collapse (DNC) in the context of binary classification. While this specific focus allows for in-depth analysis, it may limit the generalizability of the findings to other problem domains or network architectures. Further exploration of DNC in more diverse settings could strengthen the paper's applicability.

$\bf Typos:$ (1) At line 122, it should be $W_L \in \mathbb{R}^{K\times d_L}$.

**Questions:**

$\bf 1.$ The results in this paper are based on the L-layer deep unconstrained features model. This omits all the structures in the data input. A natural question is whether the analysis can be extended to the case where the data input has some special structures, such as the case where the data is whitened, i.e., assuming $H_1H_1^T=I$ in Definition 1.

$\bf 2.$ What is the main technical difficulty in analyzing multiple-class cases?

$\bf 3.$ What is the role of the activations in deep neural networks? Why do DNC1 and DNC2 hold for layer l ≥ 2 before and after activations?

---

> ### Author Rebuttal · Authors · 2023-08-09
>
> We thank the reviewer for the thoughtful assessment and for the positive evaluation of our work. We address the points raised in the review below.
>
> **Theorem 3 cannot explain the progressive neural collapse**
>
> *Response.* We agree that there is a gap between the predictions of the DUFM and the progressive neural collapse which has been observed empirically in real networks, and investigating this gap is an important future direction. Let us make two comments in this regard.
>
> *(i)* The progressivity of DNC may be due to the non-linearity of the data, which the neural network has to eliminate throughout the layers. One reason for the gradual removal of within-class variability through the layers could be the balance between the norms of the weight matrices of different layers. In fact, decreasing the within-class variability more quickly would require larger matrix norms, which is prevented by the regularization.
>
> *(ii)* A dynamic analysis of the trajectory of gradient descent could also show a progressive neural collapse: the final layers reduce the within-class variability faster than the first ones, as training progresses.
>
> Finally, we note that the phenomenon of progressive neural collapse is not entirely understood even at the experimental level: on the one hand, [8] observes a log-linear behavior in the DNC1 reduction; on the other hand, [26] observes a reduction which is no longer log-linear.
>
> **Lack of discussion on the role of activations**
>
> *Response.* This is a good point, we will expand the manuscript accordingly. Our theorem indeed reveals an intriguing insight into the role ReLU plays in the DNC. As we see from Lemma 6, the optimal intermediate layer activations, i.e. $H_l$ in our notation, are non-negative.
> This is due to the fact that the ReLU disregards any negative activation by zeroing it out. From our computations, it follows that producing strictly negative values is suboptimal for the norm of $W_l$ and, therefore, all the activations are non-negative.
>
> **The paper only focuses on binary classification**
>
> *Response.* This issue was raised by multiple reviewers. Hence, we have opted to respond to it in the global response above.
>
> **Q1: Omitting the data structure + possible structure in $H_1$**
>
> *Response.* We agree with the reviewer that the unconstrained features model removes all structure from the data. In fact, the main assumption underlying the model is that the network’s backbone eliminates any structure from layer $H_1$. However, though we consider multiple last layers, we still assume that we only look at the relative end of the network, therefore the unconstrained features model is similarly motivated as in the last-layer collapse. Going beyond unconstrained features, understanding the effect of additional structure is certainly a natural question.
>
> Note that the rank of $H_1$ under DNC1 is at most $K$, and the row-dimension of $H_1$ is $d_1$. Thus, for $H_1H_1^T=I$ to hold, we need to look at $\bar{H}_1$ (as defined in line 116) instead of $H_1$ and we need $d_1 = K$, which is rather restrictive. Hence, in general, the assumption $H_1H_1^T=I$ is not compatible with DNC1.
>
> Instead of considering an $l_2$ regularization on $H_1$, several papers (see e.g. [19, 31, 36]) assume the features to be normalized, which can be regarded as a way to add structure in the input data. Another attempt to assume *some* structure is done in [29], where the authors consider features that are only approximately unconstrained and admit an error. It is an interesting future direction to quantitatively understand the effect of such modeling assumptions on our results.
>
> **Q2: Main challenges in analyzing the multiclass case**
>
> *Response.* Please see the global response.
>
> **Q3: The role ReLU plays in our theory**
>
> *Response.* The reason why DNC1 and DNC2 hold before and after the activation is rather subtle, and it follows from the results of Lemma 6. One intuitive way of reasoning about this is as follows: ReLU disregards negative values, hence it is useless for the network to create negative features. The computations of Lemma 6 formalize this idea: the network avoids creating negative features (before the application of ReLU), and by doing it, it yields a solution with the smallest possible Frobenius norm.

---

### Author Rebuttal · Authors · 2023-08-09

We would like to thank the reviewers for the positive feedback on our work. We reply to reviews separately and we address here one point raised by all reviewers.

**The paper only considers the binary classification case. Are the results generalizable to the multi-class classification and if yes, what are the main technical challenges?**

*Response.* We regard the generalization to multiple classes as an important open problem. We discuss below two technical challenges in this generalization:

*(i)* For more than two classes, Lemma 9 is false. We provide a counterexample in Appendix B for a multi-column matrix $H$ and $L=2$. This makes it difficult to obtain the optimal value of $H_2^*$, which is crucial for the rest of the analysis. On the other hand, even if Lemma 9 does not hold in full generality, [29] argues that the inequality cannot be violated for the (supposedly optimal) orthogonal matrices $H$. This indicates that counterexamples to Lemma 9 have to be ill-conditioned, which leads to an increase in the loss also in the multi-class case. Therefore, obtaining some bounds on $||H||_*$ could suffice to generalize the proof.

*(ii)* Computing the optimal solution of Lemma 5 appears to be challenging for multiple classes. In fact, a component-wise application of the ReLU activation can increase the rank of a matrix. In other words, even if $X$ is low rank, $\sigma(WX)$ can still have high rank, which significantly complicates the analysis.

**Enclosed PDF:**

Some of the reviewers asked us to provide additional experiments supporting the validity of DUFM as a modeling principle. Therefore we provide experiments where we trained VGG13 and DenseNet121 as an alternative to ResNet20 for the 3-layer DUFM head and alternative ResNet20 training for 5-layer DUFM head. The results are in full accordance with those in the paper.

---

### Decision · Program_Chairs · 2023-09-21

**Decision:**

Accept (spotlight)

**Comment:**

This paper presents a robust theoretical analysis of deep neural collapse through the deep unconstrained features model, expanding understanding of deep network behavior during training. While it offers substantial insights and empirical backing, it could benefit from extending its theoretical scope beyond binary classification and considering various loss functions. Nevertheless, its current contributions are promising.